# Myosin-dependent short actin filaments contribute to peripheral widening in developing stereocilia

Xiayi Liao [1], Chun-Yu Tung[1], Jocelyn F. Krey[2,3], Ghazaleh Behnammanesh[4], Joseph A. Cirilo Jr. [5], Mert Colpan[6], Christopher M. Yengo[5], Peter G. Barr-Gillespie [2,3], Jonathan E. Bird [4] & Benjamin J. Perrin [1] ✉

Stereocilia, the actin-based mechanosensory protrusions of inner ear sensory hair cells, require precise dimensional control for proper mechanotransduction, yet the mechanisms governing actin assembly during development remain unclear. Their size and shape are determined by a stable core of long, parallel, unbranched filamentous (F-) actin. We find that during stereocilia widening, which is a key process for function and stability, newly expressed actin first integrates at the tip, then along the periphery of the core. To understand how actin assembles, we probe for globular (G-) actin, F-actin barbed ends, and pointed ends, and identify a tip-enriched population of short actin filaments. Overexpressing actin increases these filaments at the stereocilia periphery, suggesting a role in widening. In addition, the tip-localized myosins MYO3A/B and MYO15A, essential for normal growth, generate or stabilize short filaments. We propose that these short filaments are intermediates that mature into the long F-actin known to comprise the stereocilia core.

Stereocilia are giant microvilli-like protrusions on sensory hair cells that are specialized to detect sound as well as linear and angular acceleration in the inner ear. The mechanosensitive function of hair cells relies on stereocilia that are organized into rows of decreasing heights, which in the case of cochlear hair cells can detect nanometer-scale deflections induced by sound. Similar to microvilli, stereocilia are built around a filamentous actin (F-actin) core where hundreds of filaments are packed together and uniformly oriented with their fast-growing barbed ends towards the protrusion tip[1,2]. Regulation of the growth and stability of these F-actin bundles is crucial for stereocilia development and maintenance.

Early in development, stereocilia resemble microvilli before undergoing stages of growth where they lengthen or widen. These growth stages have been best described in mouse auditory inner hair cells (IHCs)[3,4], which are the cochlear hair cell subtype that transmits sound information via afferent innervation to the central nervous system. In stage I, microvilli emerge on the IHC apical surface, then a subset of those microvilli elongate in stage II to form stereocilia that are arranged in rows of different lengths. Stereocilia then widen during stage III, which corresponds to postnatal days 0-8 for IHCs in the apical turn of the mouse cochlea. Finally, in stage IV, stereocilia in the tallest row, referred to as row 1, elongate to their final length[4] to produce the mature stereocilia bundle with a staircase-like morphology (Fig. 1a). These observations suggest actin behaves differently in each growth stage to either lengthen or thicken the F-actin bundle, which dictates stereocilia shape and ultimately hair cell mechanosensitivity.

Several studies have documented actin behavior in developing and mature stereocilia[2,5-9], but have not converged on a clear

[1]Department of Biology, Indiana University, Indianapolis, IN, USA. [2]Oregon Hearing Research Center, Oregon Health & Science University, Portland, OR, USA. [3]Vollum Institute, Oregon Health & Science University, Portland, OR, USA. [4]Department of Pharmacology and Therapeutics, University of Florida, Gainesville, FL, USA. [5]Department of Cellular and Molecular Physiology, Penn State College of Medicine, Hershey, PA, USA. [6]Department of Cellular and Molecular Medicine, The University of Arizona, Tucson, AZ, USA. ✉e-mail: bperrin@iu.edu

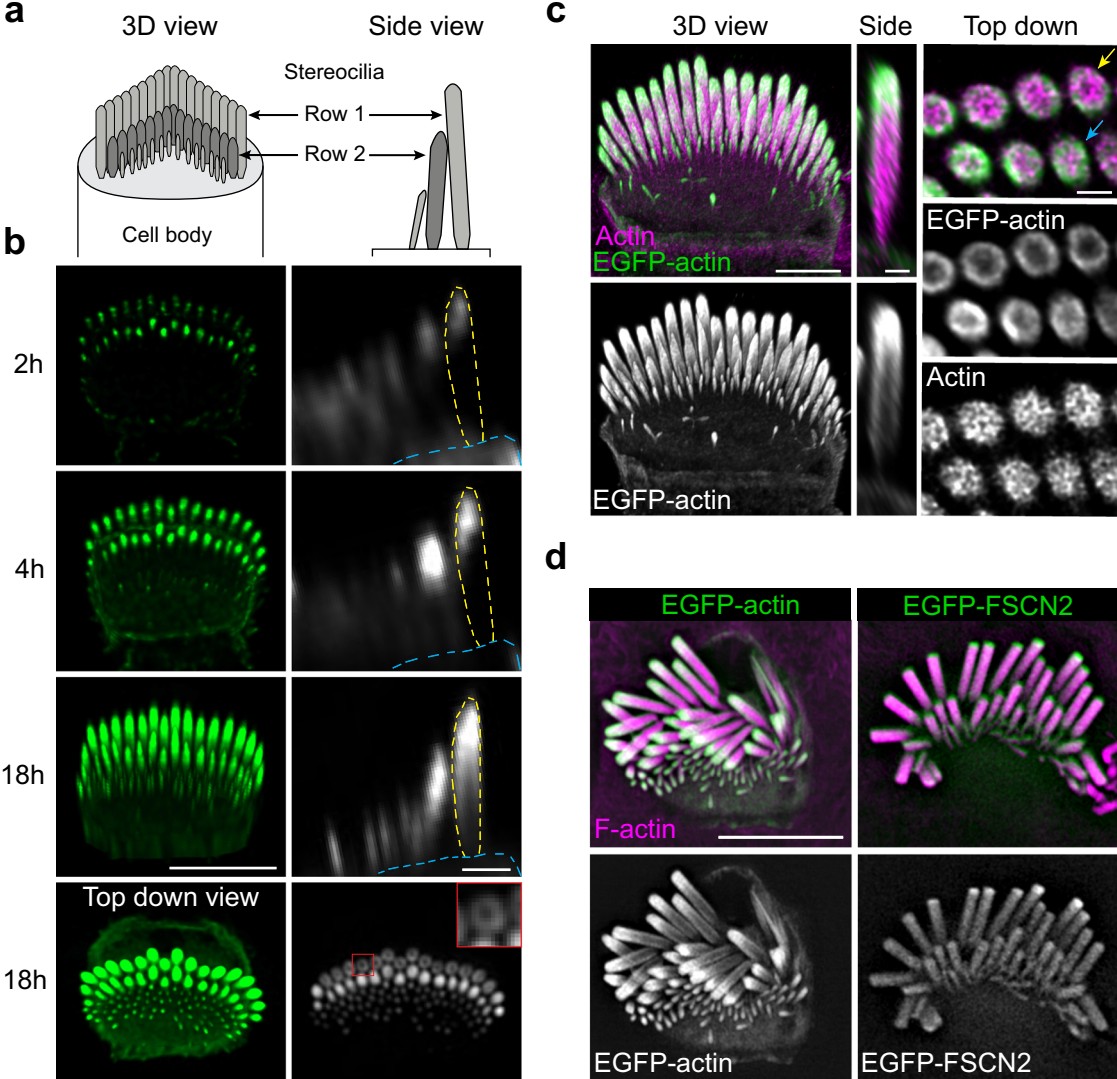

**Fig. 1 | EGFP-actin incorporation pattern during stereocilia widening.**
**a** Diagrams of inner hair cell (IHC) stereocilia, oriented *en face* in a 3D view and as a side view. **b** A single P6 IHC imaged at timepoints (from 2 to 18 h) after transfection with EGFP-actin. Left panels are 3D reconstructions oriented *en face* except for the final image, which is a top-down view (scale bar represents 5 μm). Right panels are side views of stereocilia made from x-z reslices (scale bar represents 1 μm). The outline of a row 1 stereocilium is traced by yellow dashed lines and the cuticular plate is denoted by blue dashed lines. The lowest panel is an x-y slice showing stereocilia in cross-section mid-way down row 1 stereocilium. The inset (1 ×1 μm), a magnified region marked by a red box, demonstrates peripheral EGFP-actin

localization around row 1 stereocilia. **c** P5 IHC 18 h post EGFP-actin transfection imaged by expansion microscopy, stained with antibodies against β-actin (ACTB) (magenta) and EGFP (green). Panels show a 3D reconstruction (Scale bar represents 10 μm), a x-z reslice through the center of stereocilia, and a top-down view of x-y slice (Scale bars represent 2 μm). Yellow and blue arrows indicate row 1 and row 2, respectively. **d** Lattice SIM images of EGFP-actin or EGFP-FSCN2 (green and grey) from P5 IHCs 18 h post transfection with phalloidin-stained F-actin (magenta). Scale bar represents 5 μm. **b**–**d** The experiments were repeated three times with similar results.

understanding of how actin assembles to produce the correct stereocilia core dimensions. Early studies[2,5] assessed EGFP-actin localization in fixed cells at timepoints after transfection and observed that EGFP-actin incorporated at stereocilia tips first, and then seemed to progress down the stereocilia shaft with time. These observations were interpreted as evidence of actin treadmilling, such that the stereocilia actin core underwent continuous renewal where actin monomers added at the tips and were released at the bases. Subsequent studies[6–9] using a variety of approaches confirmed that stereocilia actin incorporation is most evident at stereocilia tips, and further demonstrated that actin addition at tips can drive elongation. However, these studies also demonstrated that the stereocilia actin core was highly stable once formed, thus contradicting the treadmilling model. Absent treadmilling, it is unclear how actin assembly at stereocilia tips contributes to stereocilia widening.

Stereocilia lengthening and widening is known to depend on different tip-localized unconventional myosin complexes. MYO15A is a core component of the elongation complex, so named because IHCs lacking any member of the complex have exceptionally short stereocilia[10–14]. The elongation complex localizes to row 1 and row 2 tips in early postnatal development, and more exclusively to row 1 tips during stage IV growth. The paralogs MYO3A and MYO3B also localize to both row 1 and row 2 stereocilia tips[15–17]; however, their loss results in stereocilia that are longer and thinner than normal stereocilia[16], suggesting the myosin-III proteins promote widening at the expense of elongation. While MYO15A, MYO3A, and MYO3B play pivotal roles in stereocilia growth, it is unclear how they regulate actin behavior to promote elongation or widening. Stereocilia widening is particularly puzzling because there is not a known mechanism by which actin incorporation at stereocilia tips can lead

to the addition of new F-actin to the seemingly stable stereocilia actin core.

We investigated how stereocilia widen, first by using high-resolution live-cell imaging of transfected IHCs to visualize actin incorporation in widening stereocilia. Consistent with prior work, fluorescent actin first incorporated at stereocilia tips before extending to the stereocilia shaft. Rather than a treadmilling mechanism, the new actin surrounded the existing, stable actin core, suggesting that new filaments incorporate first at stereocilia tips and then along the stereocilia shaft to produce new, peripheral core filaments. We then used actin binding proteins as probes to reveal that G-actin, as well as F-actin barbed and pointed ends, are all detectable at stereocilia tips. Discovering F-actin pointed ends at stereocilia tips was unexpected because F-actin in the stereocilia core is thought to be uniformly oriented with only barbed ends at tips[1]. This discrepancy suggests that there is a population of short actin filaments at stereocilia tips. The level of this actin species depends on MYO15A and MYO3A/B, suggesting that short actin filaments serve as intermediates in myosin-dependent stereocilia actin assembly.

## Results

### Stereocilia widen by adding new actin around stable core

To understand how widening occurs within stereocilia, we revisited the question of EGFP-actin incorporation by imaging individual P6 IHCs at intervals after transfection using high-resolution Airyscan microscopy. In these experiments, EGFP-actin invariably incorporated at the tips of stereocilia first, and then subsequently appeared along the stereocilia shaft (Fig. 1b). In contrast to the treadmilling model[2,5], we observed that EGFP-actin added to the periphery of the stereocilia shaft, but not to the preexisting F-actin core. This pattern of addition was particularly evident 18 h after transfection in longitudinal slices showing the entire stereocilia length, and in cross sections in the middle of the stereocilia where EGFP-actin appeared as a ring (Fig. 1b, top-down view). To further define the spatial localization of transfected EGFP-actin, we also imaged samples that were fixed 18 h after transfection at higher resolution using both expansion microscopy (U-ExM) (Fig. 1c) and lattice structured illumination microscopy (lattice-SIM) (Fig. 1d). As with live cells, the most typical incorporation pattern was a bright cap of EGFP-actin at the stereocilia tip and a thin ring of EGFP-actin around the stereocilia shaft (Fig. 1c, d). Culturing P5 IHCs for 48 h after EGFP-actin transfection produced a similar, peripheral incorporation pattern, consistent with the existing core being stable once formed (Supplementary Figs. 1a, b). We also transfected P3 IHCs, and in this case EGFP-actin incorporation was also peripheral, but less evident, suggesting the stable F-actin component of the core is thinner earlier in development (Supplementary Fig. 1c, d). In contrast to EGFP-actin, transiently expressed fascin-2 actin crosslinker (EGFP-FSCN2), imaged with lattice-SIM, infiltrated into the existing core of P6 IHC stereocilia (Fig. 1d, right panel), which is consistent with a previously documented EGFP-FSCN2 turnover pattern[18] and shows that the core is accessible to proteins similar in size to actin. Together, these findings demonstrate new EGFP-actin incorporates first at the tip, then surrounds the existing stereocilia F-actin core.

### G-actin, F-actin barbed ends, and F-actin pointed ends are present at stereocilia tips

Actin incorporation at stereocilia tips is thought to result from polymerization at the barbed ends of the actin filaments that are part of the stereocilia core[2,5–7]. However, it is unclear how actin filaments are added to the periphery of the core to cause widening. We hypothesized that widening depends on an intermediate arrangement of F-actin during stereocilia development, which we sought to characterize using probes to detect G-actin, as well as the barbed or pointed ends of F-actin. We analyzed intact cells as well as cells extracted with saponin, which depletes the soluble G-actin pool from

cells[19]. To characterize the distribution of G-actin in IHC stereocilia, we first expressed RPEL1-EGFP, a G-actin binding domain from the protein MAL fused to EGFP[20], in IHCs. This G-actin reporter was detected throughout the stereocilia, but was noticeably enriched at tips (Fig. 2a, b). When cells were extracted with saponin, RPEL1-EGFP signal was drastically reduced to nearly undetectable levels (Fig. 2a, right panels). We also probed for G-actin in freshly dissected, saponin-permeabilized cochlear tissue from P5 mice with the JLA20 antibody, which selectively binds to G-actin but not F-actin[19]. Under these conditions JLA20 staining was enriched at row 2 tips but was not evident at row 1 tips (Fig. 2c–e). In contrast, the AC-15 anti-β-actin antibody[19] stained the entire length of stereocilia (Fig. 2c, left panels). Thus, JLA20 staining likely marks a population of G-actin at row 2 stereocilia tips that is anchored to prevent extraction by saponin treatment. Interestingly, the JLA20 signal was absent by P9 (Fig. 2c, e), suggesting that G-actin at row 2 tips is developmentally regulated.

To further define the actin network in stereocilia, we transfected cells with constructs encoding the F-actin binding proteins EGFP-TMOD1 and EGFP-SH3BGRL2, which each recognizes F-actin pointed ends[21,22]. Both pointed-end binding proteins preferentially bound to stereocilia tips compared to the stereocilia shaft (Fig. 3a, b, Supplementary Fig. 2a, b). Following saponin extraction, the tip enrichment of TMOD1 signals became more evident, with a greater reduction in shaft signal (Fig. 3a, c), indicating that a subpopulation of TMOD1-bound F-actin is anchored at tips. To further assay for F-actin pointed ends, we probed saponin permeabilized cochlear tissue with purified actin pointed-end binding proteins DNaseI[23] and His-TMOD1. Both proteins bound robustly at row 1 and row 2 stereocilia tips (Fig. 3d). We detected F-actin barbed ends with a similar strategy, this time employing purified His-CAPZ protein[24]. As expected, based on the known arrangement of F-actin core filaments, His-CAPZ preferentially labeled the tips of stereocilia (Fig. 3d). While F-actin barbed ends are expected at stereocilia tips because actin in stereocilia is comprised of parallel filaments with barbed ends terminating at tips, detecting pointed ends at tips was unexpected. The pointed end signal suggests that there is a population of short actin filaments at stereocilia tips (Fig. 3e). We will refer to these putative actin filaments as tip filaments to distinguish them from the bundled core filaments, which comprise the main structure of stereocilia.

### Orientation, solubility and dynamics of tip filaments

To better understand the relationship between barbed and pointed end probes at stereocilia tips, we compared the relative localization or offset of maximal value of the His-CAPZ, His-TMOD1, or DNaseI spot from the tip of the phalloidin-stained F-actin in stereocilia in lattice-SIM images (Fig. 3f, g). The peak intensity of these probes from a histogram (Fig. 3h) had similar displacement for the pointed-end probes DNaseI and His-TMOD1. The barbed end probe, His-CAPZ, was more distal, indicating that the barbed and pointed ends are spatially separated. These results are consistent with a model where short tip filaments run parallel with core filaments such as that presented in Fig. 3e.

If tip filaments are separate from stereocilia core filaments, then they should have different solubility and behavior. To test this idea, we extracted freshly dissected cochlear tissue with a high-salt buffer to disrupt protein binding interactions that depend on electrostatic charge, and then labeled His-TMOD1 in standard buffer. The phalloidin-stained stereocilia core appeared unchanged by high-salt extraction but pointed end labeling at stereocilia tips was reduced by around 70% (Supplementary Fig. 3a, b), which is consistent with extraction of tip filaments. Barbed end levels probed by His-CAPZ were also reduced (Supplementary Fig. 3c, d), though to a lesser extent, likely reflecting labeling of the remaining core filament barbed ends. To assess tip filament behavior, we incubated postnatal cochlear

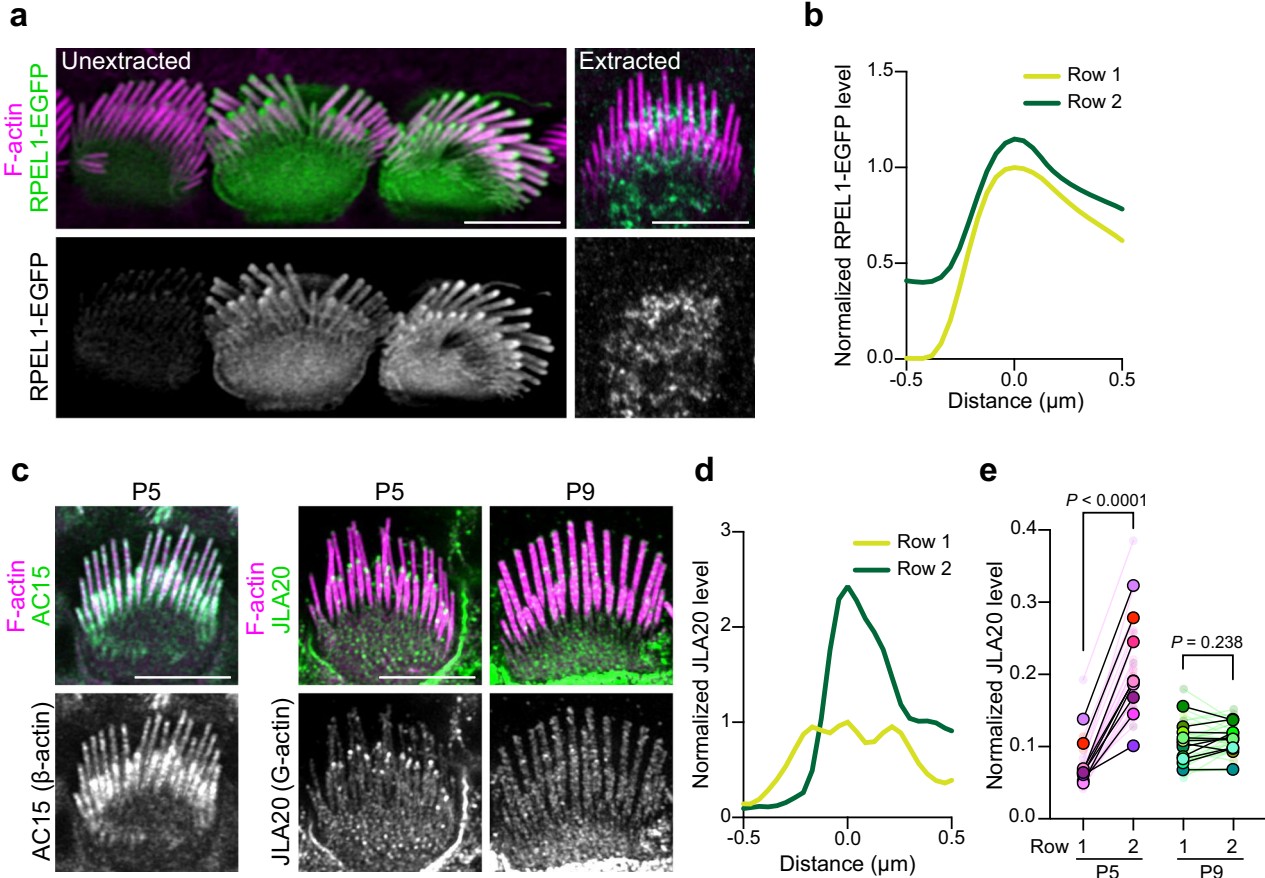

**Fig. 2 | G-actin is enriched at stereocilia tips. a** IHCs 18 h post transfection with RPEL1-EGFP, unextracted or extracted by saponin before fixation. F-actin was stained with phalloidin (magenta). The experiments were repeated three times with similar results. **b** Representative line scans drawn down the center of stereocilia in RPEL1-EGFP transfected, unextracted IHCs at P5. In (**b**) and (**d**), line scans were drawn from tip towards the base down the center of stereocilia, the peak fluorescent signal was set as 0 on x axis, and the fluorescent intensity was normalized to the row 1 maximum. **c** Images of IHCs from saponin-permeabilized P5 or P9 mouse organ of Corti that was probed before fixation with antibodies against β-actin (AC15) or G-actin (JLA20) (green and grey); F-actin was counterstained post-fixation with phalloidin (magenta). **d** Representative line scans of JLA20 stained P5 IHC stereocilia. **e** Quantification of JLA20 level at stereocilia tips normalized to the average intensity of cell junctions. Sample size (cells, cochleae): P5 (24, 9), P9 (33, 11). Smaller circles represent the average value of stereocilia tips from individual cells. Larger open circles represent the average value of cells from one individual cochlea (N). P values from two-tailed paired t tests based on N are indicated. $P < 0.05$ is considered statistically significant. Scale bars represent 5 μm. Source data are provided as a Source Data file.

explants in media containing latrunculin A (LatA), a drug that sequesters G-actin and depolymerizes dynamic F-actin structures[25]. In P5 explants, LatA treatment for 1 h reduced His-TMOD1 labeling by 50% at row 1 tips and 65% at row 2 tips. Pointed end levels largely recovered 4 h after the drug was washed out (Fig. 4a, b). The loss of tip filaments with LatA treatment, and their regeneration following washout, show that they require ongoing polymerization to stay intact, which is behavior typical of dynamic actin networks.

### Tip filaments are more abundant at widening stages of stereocilia growth

If tip filaments contribute to stereocilia growth, their levels would likely correlate with when stereocilia are widening during stage III growth (P0-P8) or elongating during stage IV (P8-P15)[4]. By probing permeabilized cochlea dissected from mice of postnatal ages, we found that His-TMOD1 staining at tips was highest at P3, and at that age the staining was similar at row 1 and row 2 tips. As development progressed, staining decreased at the tips of both rows, but more rapidly in row 2 so that by P9 signal was faint at row 1 tips and lost at row 2 tips (Fig. 4c–e). His-TMOD1 labeling of stereocilia shafts, though faint compared to staining at tips, followed the same trend, decreasing between P7 and P8 as widening slowed (Fig. 4c, f). Thus, pointed end levels at stereocilia tips and along the shaft correlated well with

widening, but tip filaments were still present at row 1 stereocilia tips when they switched to elongation.

### Discontinuous F-actin formed when stereocilia widen by actin overexpression

To assess the actin states during widening, we overexpressed EGFP-actin and measured the level of barbed and pointed ends in the shaft as stereocilia widen (Fig. 5). Actin overexpression increased stereocilia width compared to untransfected cells as assessed by measuring the full width at half maximum intensity of phalloidin staining (Fig. 5b, Supplementary Fig. 4a). His-TMOD1 and His-CAPZ staining of saponin-permeabilized tissue was increased in stereocilia shafts of EGFP-actin expressing cells (Fig. 5c, d). The staining was more enriched at the periphery of the shafts (Supplementary Fig. 4b–d), corresponding to the addition of EGFP-actin added to the sides of the existing F-actin core. As both barbed and pointed ends were detected uniformly along the stereocilia length, the newly added actin is likely composed of short filaments, rather than long filaments that grow continuously from either the stereocilia tip or base.

To assess the requirement of barbed or pointed end polymerization for actin incorporation at the tip or shaft, we expressed red fluorescent protein (RFP)-tagged polymerization-incompetent mutants DVD-actin (D286A, V287A, D288A) and AP-actin (A204E,

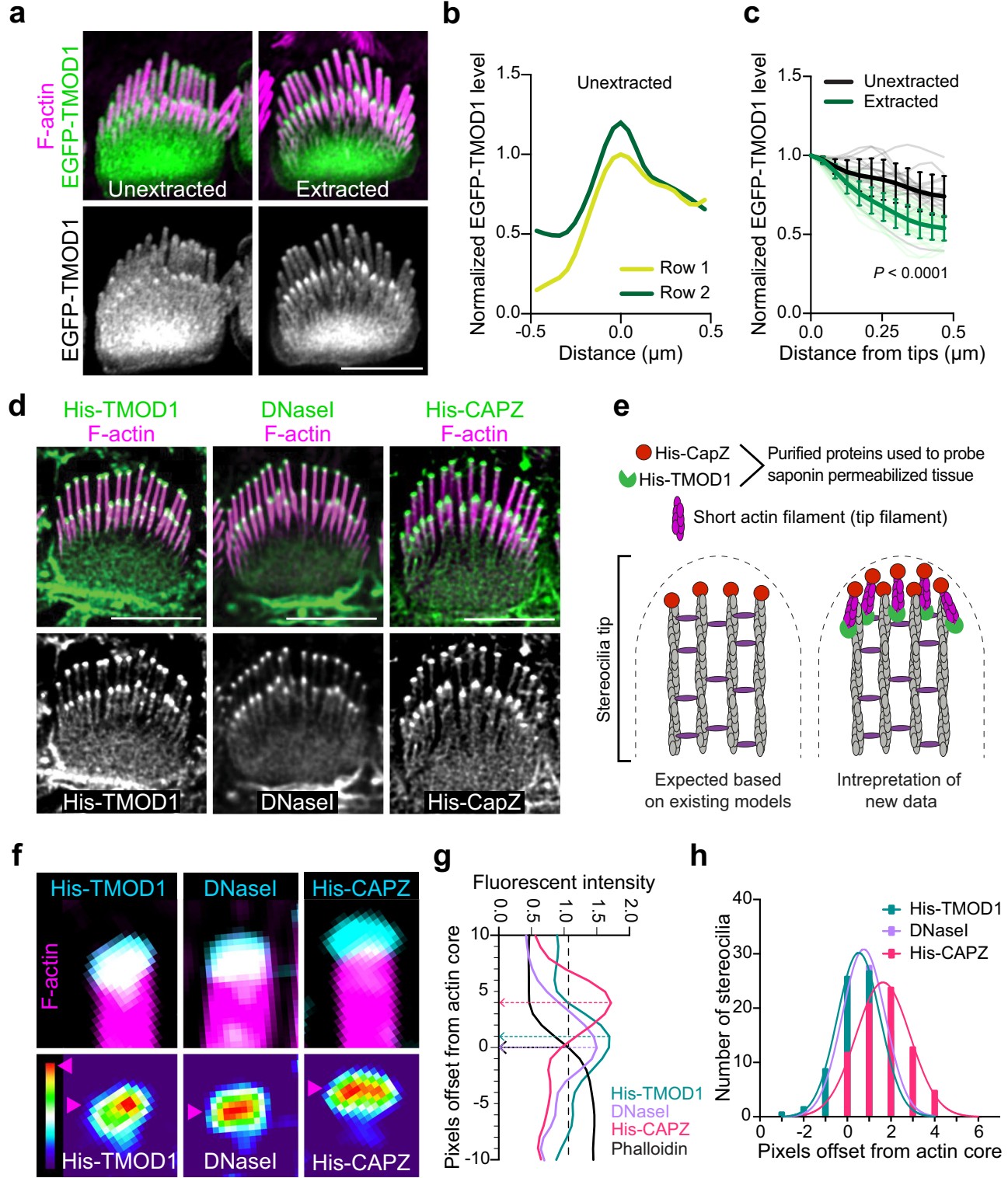

P243K). Existing actin filaments can incorporate DVD-actin at their barbed ends and AP-actin at their pointed ends, but the mutant actin then blocks subsequent monomer addition[26–28]. Both mutant actins localized to stereocilia tips, but did not appreciably incorporate along the stereocilia shaft as compared to neighboring cells that expressed normal EGFP-actin (Fig. 6a). Interestingly, in co-expression experiments, AP-actin, and to a lesser extent DVD-actin, decreased co-expressed EGFP-actin incorporation in the tip region (Fig. 6a, b). In contrast, EGFP-actin still incorporated similarly along the shaft whether or not it was co-expressed with mutant actin, which is evident in

longitudinal line scans along stereocilia (Fig. 6c). These observations suggest that actin incorporation is regulated differently at the stereocilia tip than along the shaft.

**Tip filament levels vary with MYO3 and MYO15 expression**
Our data suggests that short actin filaments at stereocilia tips and along the shaft are key intermediates in forming the stereocilia core. We next wanted to understand how tip filaments relate to tip-localized myosins, which regulate stereocilia elongation and widening. Two different classes of myosins, including MYO3A and MYO3B, encoded

**Fig. 3 | F-actin barbed ends and F-actin pointed ends are present at stereocilia tips. a** EGFP-TMOD1 transfected P5 IHCs, either unextracted or extracted by saponin before fixation, that were also stained with phalloidin for F-actin (magenta). **b** Representative line scans drawn down the center of stereocilia of EGFP-TMOD1 transfected, unextracted P5 IHCs. The peak EGFP level was set as 0 on x axis and the fluorescence intensity was normalized to the maximal fluorescence intensity of row 1. **c** Line scans quantifying EGFP-TMOD1 levels from stereocilia tips toward shafts in saponin-extracted or unextracted P5 IHCs. Sample size (stereocilia, cells, cochleae): unextracted (18, 6, 2), extracted (33, 11, 3). The shadow lines represent individual stereocilia; the thick solid lines are the average level of all stereocilia. The data were plotted as mean ± SD and analyzed by two-way ANOVA ($P < 0.0001$ for the effects of extraction, distance from stereocilia tips, and interactions between these parameters, based on stereocilia). **d** His-TMOD1, DNaseI, or His-CAPZ (green, grey) localization after probing permeabilized IHCs at P5-6. F-actin was stained with phalloidin (magenta). The experiments were repeated three times with similar results. **e** Diagrams of potential actin structures in stereocilia with F-actin barbed and pointed ends bound by His-CAPZ and His-TMOD1,

respectively. **f** Lattice SIM images of His-TMOD1, DNaseI, and His-CAPZ (cyan, thermal lookup table) at row 1 stereocilia tips with phalloidin-stained F-actin (magenta), which were quantified in (**g, h**). Magenta arrowheads denote the position of peak fluorescence intensity. **g** Representative line scans drawn down the center of stereocilia showing the intensity of His-TMOD1 (blue), DNaseI (purple), His-CAPZ (red), and F-actin actin (black). The stereocilia tip is indicated by black dashed arrow. Peak intensities for His-TMOD1, DNaseI, and His-CAPZ are indicated by colored dashed arrows. The offsets of the probe centers from the stereocilia tips were determined and plotted in (**h**). **h** A frequency histogram showing the pixel offsets of His-TMOD1 (blue), DNaseI (purple), and His-CAPZ (red) from the stereocilia tip. Sample size (stereocilia, cells, cochleae) of each probe (75, 15, 3). The histogram of each probe is fitted with a Gaussian curve. Mean offsets for peak of the Gaussian curves based on the pixel size (31 nm x 31 nm): His-TMOD1, 16 nm; DNaseI, 24 nm; His-CAPZ, 51 nm. R-squared value of the fit: His-TMOD1, 0.996; DNaseI, 0.976; His-CAPZ, 0.982. Scale bars represent 5 µm. Source data are provided as a Source Data file.

by the *Myo3a* and *Myo3b* genes, as well as MYO15A, localize to stereocilia tips[10,15]. MYO15A is essential for normal elongation and for establishing the unique protein complements that define row 1 and row 2 tips[10,11,29]. MYO3A/B, in contrast, is required to establish normal stereocilia dimensions, which are abnormally tall and thin in the mouse double knockout[16,17]. As an initial test whether myosins interact with tip filaments, we incubated saponin-permeabilized cochlear tissue with 4 mM sodium orthovanadate. This inhibitor stabilizes ADP-Pi bound myosin, which is a conformation that binds F-actin weakly[30,31]. Compared to the control condition, vanadate treatment decreased the level of His-TMOD1 staining at stereocilia tips and along the stereocilia shaft (Supplementary Fig. 5a, b). This reduction is consistent with the hypothesis that myosins normally bind tip filaments and prevent their loss following cell permeabilization.

We next sought to localize tip filaments relative to MYO15A and MYO3A which are known to occupy different zones at stereocilia tips. Using U-ExM and lattice SIM, we found tip filaments overlapped both the broader MYO3A zone and the more distal MYO15A zone (Fig. 7a, Supplementary Fig. 5c, d). The specific patterns differed according to stereocilia row. In row 1, MYO15A localized as a cap at stereocilia tips; by contrast, MYO3A localized just below the MYO15A zone. Tip filaments, labeled with His-TMOD1 detected by an anti-TMOD1 antibody, were detected in both the MYO15A and MYO3A zones (Fig. 7a). In row 2 stereocilia, MYO15A antibody staining was reduced compared to row 1 and was restricted to a small patch at the distal tip (Fig. 7a, right panels). MYO3A staining was distributed more broadly at the tip (Fig. 7a, middle panels). As at row 1 tips, His-TMOD1 labeled tip filaments overlapped both myosins, but most tip filaments in row 2 coincided with the more prevalent MYO3A staining.

To assess the consequence of myosin loss on tip filaments in IHC stereocilia, *Myo3a* and *Myo3b* were mutated with CRISPR/Cas9 that was delivered to E0.7 embryos in situ by the i-GONAD technique[32], which generated a variety of small deletions in early exons of each gene that encode N-terminal kinase domains (Supplementary Fig. 6a). *Myo3a*$^{\Delta14/\Delta14}$ *Myo3b*$^{\Delta12/\Delta12}$ IHCs in mice bred from a G₀ founder (mutant 2 alleles, Supplementary Fig. 6a) had long and thin IHC stereocilia at P4 that resembled those from previously described *Myo3a/b* double knockouts[16] (Fig. 7b). The mutant IHC stereocilia had markedly reduced His-TMOD1 staining at row 1 and row 2 stereocilia tips compared to cells from wild-type tissue, or from littermate mice heterozygous at both loci (Fig. 7b, c). Using a similar approach, we also generated a *Myo15a* mutant mouse with a 25-base pair deletion in exon 19, which encodes the motor domain. Homozygous *Myo15a*$^{\Delta25/\Delta25}$ IHCs had reduced pointed-end labeling as compared to heterozygous littermates (Fig. 7d-e), though the decrease was less than for the *Myo3a/b* mutants. A similar reduction in His-TMOD1 staining was observed in mice homozygous for the well-characterized *Myo15a shaker* allele

(Supplementary Fig. 6d), which has a mutation in the motor domain thought to abolish the function of all isoforms[11,13,29,33]. In addition, one G₀ CRISPR/Cas9 *Myo15a* mutant had a mosaic phenotype where hair cells with normal morphology and His-TMOD1 staining were adjacent to cells with short stereocilia and reduced His-TMOD1 levels (Supplementary Fig. 6c). Finally, MYO3A localized normally in the *Myo15a*$^{\Delta25/\Delta25}$ mutant (Supplementary Fig. 6e). Similarly, MYO15A localized to stereocilia tips at normal levels in the *Myo3a/b* double mutant (Supplementary Fig. 6f), suggesting that each myosin independently contributes to tip filament levels.

To determine if increasing MYO15A or MYO3A protein levels increased tip filament levels, we transfected IHCs with EGFP-MYO3A (Supplementary Fig. 7a); EGFP-K50R-MYO3A, which lacks kinase activity that autoinhibits motor function[34] (Fig. 7f); or EGFP-MYO15A isoform 2 (Fig. 7h). His-TMOD1 staining at both row 1 and row 2 tips increased with EGFP-MYO3A expression (Supplementary Fig. 7a) or with EGFP-K50R-MYO3A levels (Fig. 7f, g). EGFP-MYO15A-2 expression also increased His-TMOD1 staining, albeit primarily at row 1 tips, reflecting the localization pattern of this myosin (Fig. 7h, i). Of note, stereocilia were slightly wider with overexpression of MYO3A (Supplementary Fig. 7b, c), which was not observed with MYO15A transfection, suggesting that tip filaments connected to MYO3A are more likely to enter the widening pathway. Together, these data show that the level of short actin filaments at stereocilia tips is influenced by both MYO3A and MYO15A, suggesting that these myosins may regulate stereocilia growth in part through the use of short actin filaments.

## Discussion

This study offers insights into how actin behaves in developing stereocilia that are widening, which is a key developmental step in stereocilia maturation. Super-resolution imaging revealed that newly expressed EGFP-actin first accumulates at stereocilia tips before incorporating along the stereocilia shaft. Critically, the new actin does not replace the existing stereocilia F-actin core, but rather it surrounds it, suggesting that a stereocilia shaft widens by adding new actin filaments to its periphery. We found that the arrangement of actin at the stereocilia tip is more complicated than is accounted for by existing models, all of which describe only termination of the long F-actin filaments that form the stereocilia core. Besides these core filaments, we also characterized a dynamic population of short actin filaments at stereocilia tips using probes detecting F-actin barbed and pointed ends. Short actin filaments seem to also be involved in stereocilia widening because both barbed and pointed ends of F-actin increase along the stereocilia shaft as EGFP-actin incorporates. Finally, tip filament levels are regulated by MYO3A and MYO15A proteins, which are critical determinants of stereocilia width and length. Based on the relationship between tip filaments, myosins, and stereocilia growth, we

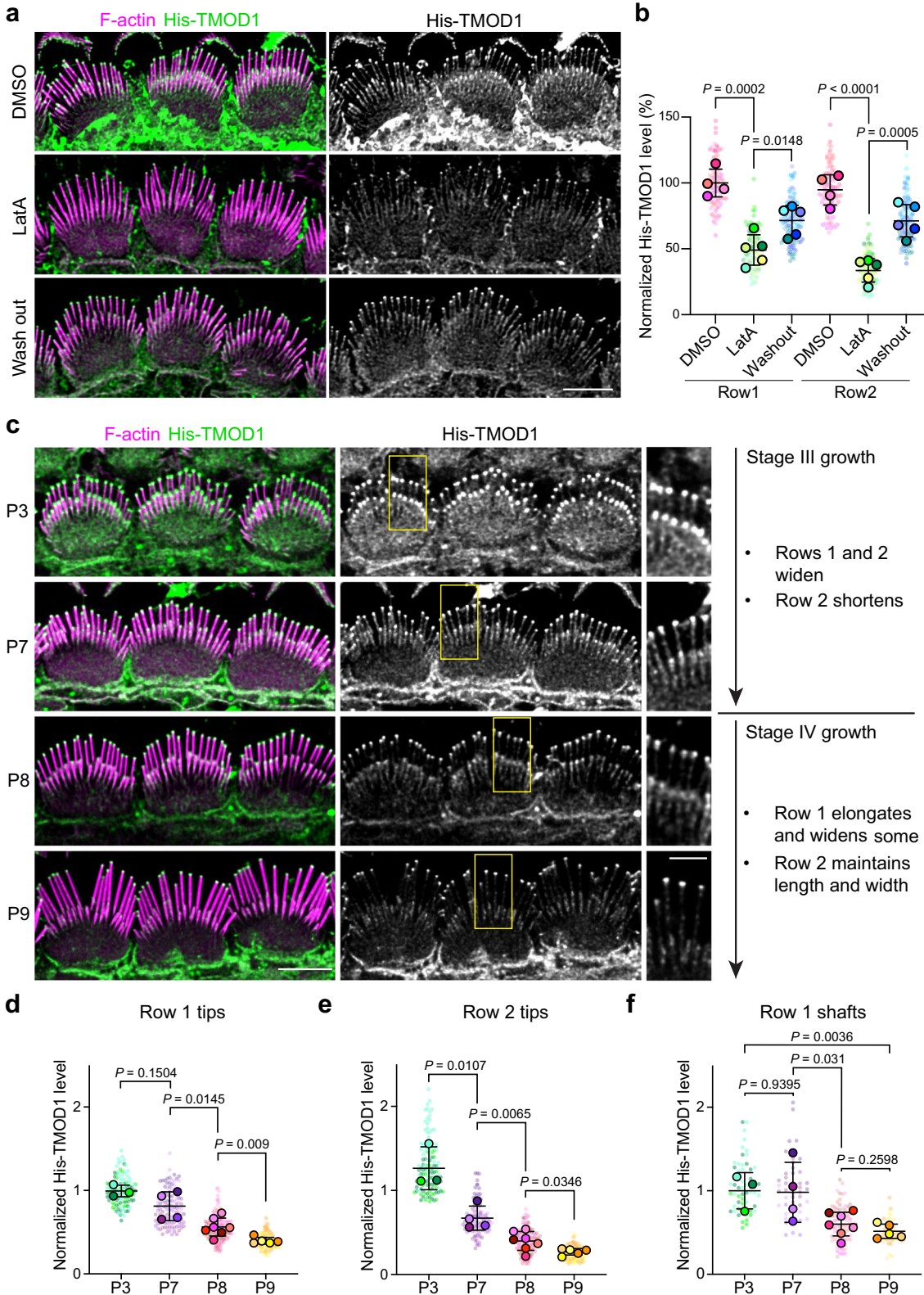

propose that myosins use, and perhaps generate, tip filaments as intermediates in the assembly of new stereocilia core filaments (Fig. 8).

### Where do tip filaments come from?

The source of short actin filaments is one of the most interesting unsolved mysteries. One possibility is that there is an as-yet unidentified nucleator, such as the ARP2/3 complex or formins, that resides at stereocilia tips and generates short filaments. Alternatively, existing core filaments could be severed by ADF or cofilin, both of which localize to row 2 stereocilia tips and regulate stereocilia growth. In addition, as levels of MYO3 and MYO15A proteins correlate with tip filaments abundance, perhaps myosins themselves directly nucleate F-actin. There is experimental support for this idea, as non-muscle myosin II has long been known to nucleate actin in vitro[35–37]. More recently, purified, recombinant MYO15A was shown to have a similar nucleation activity, both in bulk assembly assays and more directly by

**Fig. 4 | Tip filament levels during development of apical IHCs. a** His-TMOD1 staining (green, grey) of P5 IHCs after 1-hour latrunculin A (LatA) treatment and 4-hour recovery following washout; F-actin stained with phalloidin (magenta). **b** Quantification of His-TMOD1 level from row 1 and row 2 stereocilia tips. Sample size (stereocilia, cells, cochleae): DMSO (100, 20, 4), LatA or washout (125, 25, 5). The fluorescence intensity was normalized to the average fluorescence intensity of row 1 stereocilia from DMSO-treated samples and plotted as mean ± SD based on cochleae. Scale bar represents 5 µm. **c** Left panels are His-TMOD1 labeling (green, grey) of apical IHCs at P3, 7, 8, and 9. F-actin in stereocilia was stained with phalloidin (magenta). Scale bar represents 5 µm. Regions marked by yellow boxes are magnified to the side (Scale bar represents 2 µm). The timeline on the right is of IHC bundle development in mouse cochlea showing the growth details of specific rows

during stage III and IV[4]. **d, e** Quantification of His-TMOD1 level from stereocilia tips at P3, 7, 8, and 9. Sample size (stereocilia, cells, cochleae): P3 (119, 17, 3), P7 (112, 16, 4), P8 (189, 27, 7), P9 (126, 18, 5). Row 1 and 2 are separately plotted. **f** Quantification of His-TMOD1 level from row 1 stereocilia shafts at P3, 7, 8, and 9. Sample size (stereocilia, cells, cochleae): P3 (57, 19, 3), P7 (48, 16, 4), P8 (81, 27, 7), P9 (54, 18, 5). Fluorescence quantifications from (**d**) to (**f**) were normalized to the average fluorescence intensity of the row 1 level from P3 samples and plotted as mean ± SD based on cochleae. In (**b**) and (**d**–**f**), Normalized His-TMOD1 level from individual stereocilia were plotted as small dots with the color corresponding to their cochleae, represented as larger open circles. *P* values from two-tailed unpaired *t* tests comparing cochlea averages are indicated. *P* < 0.05 is considered statistically significant. Source data are provided as a Source Data file.

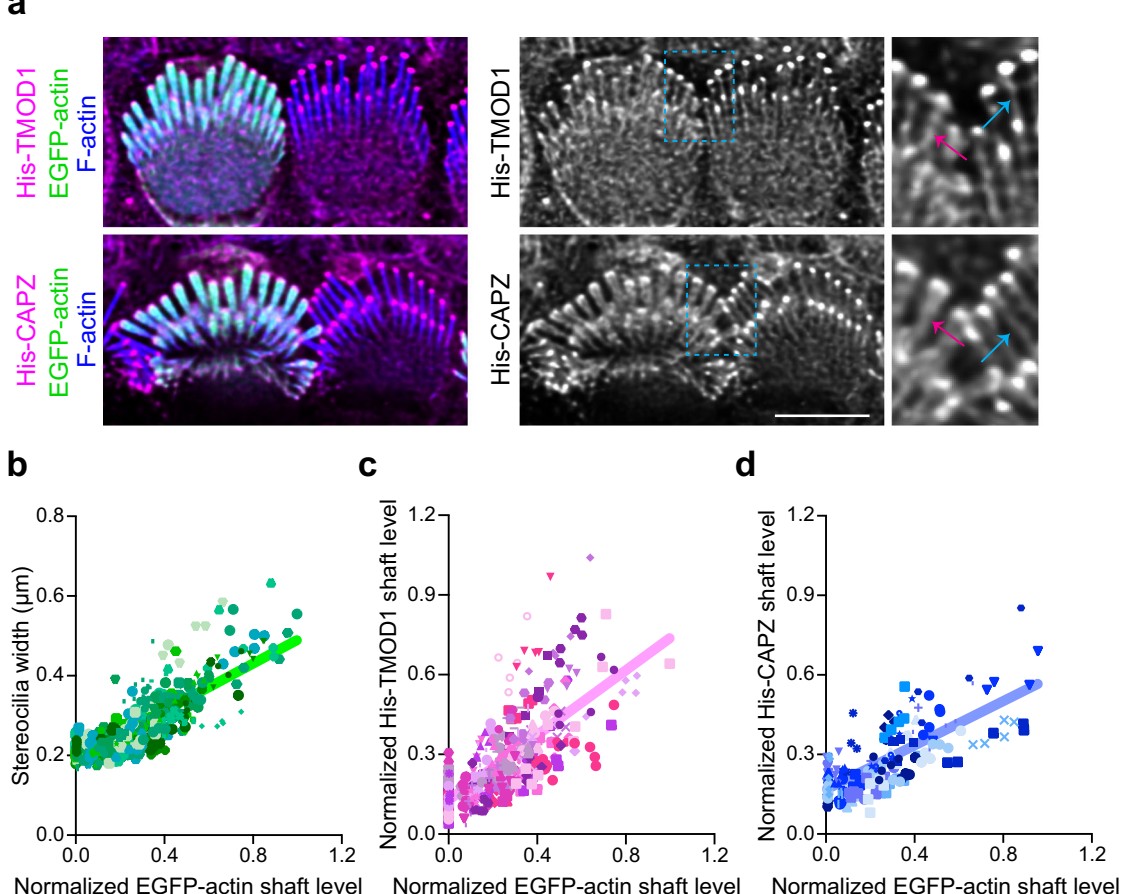

**Fig. 5 | Stereocilia widening correlates with increased F-actin barbed and pointed levels in the shaft. a** His-TMOD1 or His-CAPZ (magenta, grey) labeling in permeabilized IHCs 18 h after transfection with EGFP-actin (green). F-actin is stained with phalloidin (blue) to show stereocilia. Regions of interest are denoted by light blue dashed boxes and magnified to the right. For comparison, stereocilia shaft labeling of His-TMOD1 or His-CAPZ is indicated by blue arrows in untransfected cells and magenta arrows in EGFP-actin transfected cells. **b**–**d** Graphs showing the linear correlation of the EGFP-actin level in the stereocilia shaft with

stereocilia width (**b**), His-TMOD1 shaft staining (**c**), and His-CAPZ shaft staining (**d**). Stereocilia are plotted as individual symbols and those from the same cell are represented by identical color and shape. Simple linear regression analysis was applied to the data. R-squared values for linear regressions from (**b**) to (**d**) are 0.66, 0.52, 0.55, respectively. Sample size (stereocilia, cells, cochleae): Width (592, 119, 22), His-TMOD1 (420, 84, 13), His-CAPZ (172, 35, 9). Scale bar represents 5 µm. Source data are provided as a Source Data file.

watching filaments form in TIRF assays[38,39]. In addition, a novel *Myo15a* point mutation in the motor domain both decreased stereocilia growth and the nucleation ability of the purified protein in vitro[39]. Although a direct nucleation mechanism is intriguing because it would neatly couple the myosins that are critical for stereocilia growth with F-actin assembly, other mechanisms are conceivable. For example, myosin motors can exert enough force to break actin filaments[40] and could perhaps generate tip filaments from core filaments in this fashion. Yet another possibility is that myosins deliver a nucleation factor,

although this seems less likely considering that MYO15A and MYO3A both increase tip filament levels even though their tails bind different cargos.

### Role of short actin filaments in stereocilia widening
We propose that during stereocilia widening, short actin filaments exist along the stereocilia shaft, which subsequently mature into long actin filaments, which are well-known to characterize the actin core. This proposal is supported by His-TMOD1 staining observed

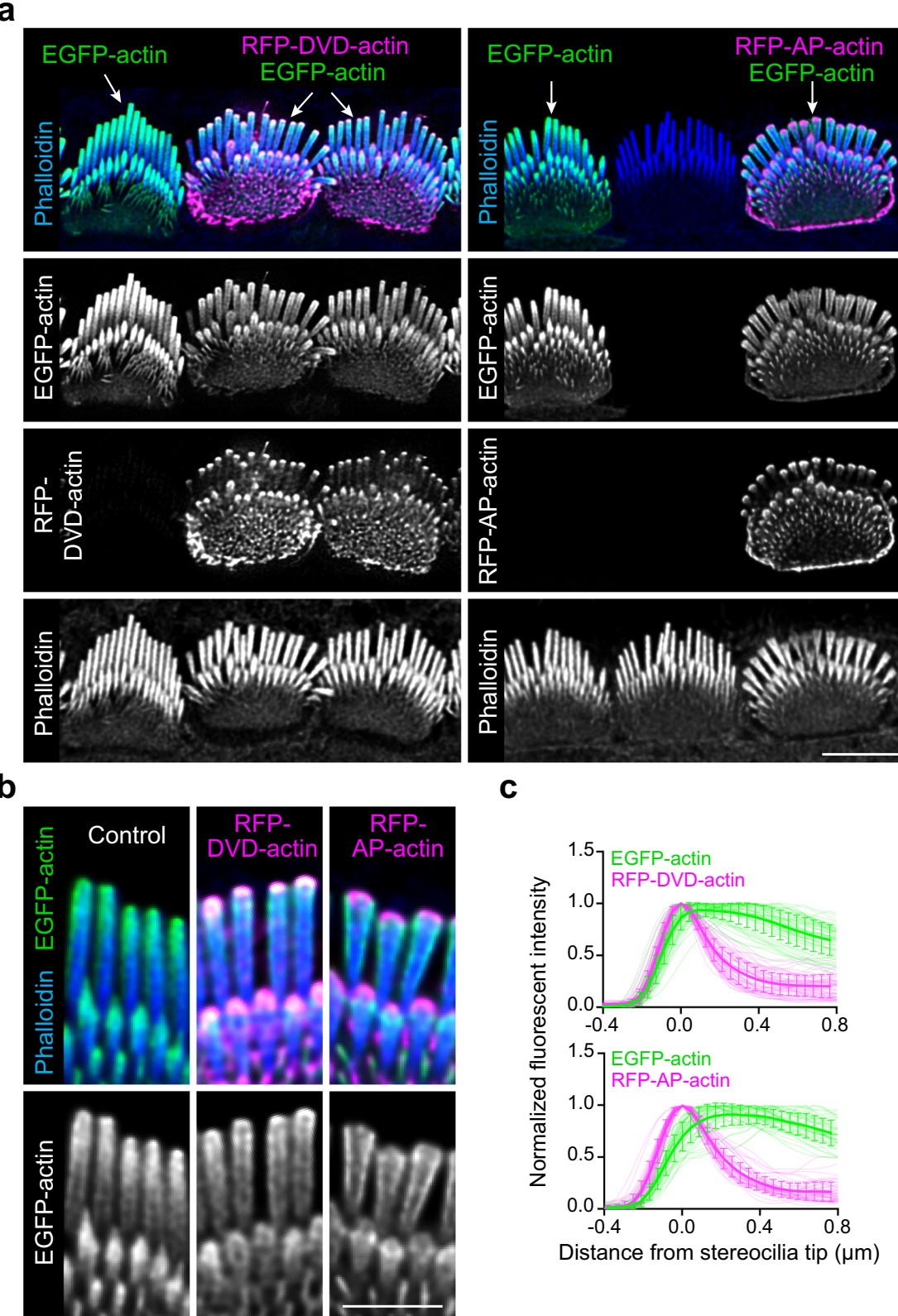

along the stereocilia shaft in stage III inner hair cells, suggesting that pointed ends are distributed throughout the length of the developing stereocilia core. Consistent with short filaments maturing into long filaments as widening concludes, pointed-end staining decreased as the cells entered stage IV. In addition, over-expressing EGFP-actin increased both barbed and pointed end labeling along stereocilia shafts, indicating that short actin

filaments were formed as stereocilia widened. Short actin filaments that contribute to widening could originate at the stereocilium tip and then move down in some fashion to populate the shaft. Such a tip origination model is consistent with data showing that actin assembly is most evident at stereocilia tips, and that MYO3A/B localizes to stereocilia tips, regulates tip filament levels, and is required for normal widening.

**Fig. 6 | Polymerization-incompetent mutant actin disrupts the incorporation of wild-type actin at stereocilia tips. a** IHCs 18 h after transfection with EGFP-actin (green) and mutant actins (magenta). Left panels: IHCs transfected with EGFP-actin alone or in combination with RFP-DVD-actin; Right panels: IHC transfected with EGFP-actin alone or in combination with RFP-AP-actin, adjacent to an untransfected IHC. White arrows denote the transfected constructs; selected regions are magnified in (**b**). Scale bar represents 5 μm. **b** Magnified insets from left to right: expression of EGFP-actin only, co-expression of EGFP-actin with RFP-DVD-actin or RFP-AP-actin. Scale bar represents 1 μm. **c** Line scans quantifying the fluorescence distribution of RFP-mutant actin relative to EGFP-actin from the co-transfected IHCs. The peak RFP level was set as 0 on x axis and the fluorescence intensity was normalized to the maximal fluorescence intensity of row 1. The line scan results were collected from −0.4 μm (above tips) to 0.8 μm (below tips) on x axis as described. The shadow lines represent individual stereocilia; the thick solid lines with error bars are the average level of all stereocilia with SD. Sample size (stereocilia, cells): DVD-actin (57, 19), AP-actin (48, 16). Source data are provided as a Source Data file.

While the tip origination model seems feasible, it is also possible that the stereocilia shaft and tip both use short actin filaments, but that those filaments are nucleated independently by distinct mechanisms. In keeping with this idea, expressing mutant, non-polymerizable actin that blocked either the barbed or pointed ends of existing filaments perturbed the addition of EGFP-actin to the tip but not to the shaft. Regardless of their origin, both models posit that short actin filaments, seeded along the periphery of an existing filament bundle, grow by monomer addition or annealing to form the long, unbranched actin filaments characteristic of stereocilia.

### The many states of actin in developing stereocilia

Identifying actin states within stereocilia is complicated by the exceptionally high density of F-actin, as well as the high concentration of other proteins at stereocilia tips, which together interfere with direct imaging approaches. We instead deployed a repertoire of G- and F-actin binding proteins as selective probes to detect either monomeric actin or the barbed or pointed ends of F-actin. Abundant data on actin structure as monomers, filaments, and filaments bound to CAPZ and TMOD1 are available and are useful for interpreting the results of our labeling experiments. At the barbed end of actin filaments, monomers present subdomains 1 and 3 and recent cryo-EM data show that these domains undergo a conformational change to flatten relative to each other as monomers polymerize onto the filament end[41]. Thus, a barbed end binding protein like CAPZ or an antibody like JLA20 can readily distinguish F-actin from G-actin. The pointed ends of the filament do not change as dramatically compared to the monomeric state[41]. Consequently, proteins like DNaseI bind to both monomers and pointed ends with high affinity. Nevertheless, some members of the tropomodulin protein family, including TMOD1, can block pointed end polymerization without binding G-actin[42]. The specificity of TMOD1 most likely arises from multivalent interaction of TMOD1 with both actin subunits that are found at the pointed end. Similarly, SH3BGRL2, a more recently identified pointed end binding protein, binds between the interface of two actin units, with no evidence that it binds to G-actin[22]. In addition to structural considerations, G-actin and F-actin are also extracted from cells differently, with G-actin largely removed from cells after treatment with low concentrations of saponin[19]. In contrast, F-actin is mostly unchanged after extraction, presumably because the polymerized actin population is large and interconnected. In the current study, we noted clear differences between probes for G-actin and for pointed ends in intact cells, which demonstrated that the pointed-end probes are not just detecting G-actin. In addition, saponin extraction nearly eliminated signal from the RPEL1-EGFP G-actin probe, while EGFP-TMOD1 was depleted from the cell body and stereocilia shaft, but not from the tip. Thus, the idea of pointed ends, and thus short actin filaments, being at stereocilia tips and along the stereocilia shaft relies on the combined results of multiple probes in intact and extracted cells.

### G-actin at stereocilia tips

Actin polymerization is most evident at stereocilia tips and, correspondingly, we found that G-actin detected by the RPEL1-EGFP probe is also enriched at stereocilia tips. It is unclear if G-actin is trafficked to tips or if it is enriched by trapping mechanism after diffusing to tips, but the RPEL1-EGFP signal is almost always absent after a short extraction with saponin. The RPEL1 protein has a relatively low affinity for G-actin, so some of the signal decrease could be dissociation of the probe. However, G-actin is well-known to be diffusible so it is more likely that the pool of G-actin at stereocilia tips is not tightly associated with existing F-actin structures or other bound proteins. This view is supported by JLA20 staining, which is not evident at row 1 tips after saponin extraction suggesting that the pool of actin marked by RPEL1-EGFP was lost.

Interestingly, JLA20 did stain the tips of row 2 stereocilia from P5 mice, but not P9 mice, after saponin treatment, revealing that some G-actin is resistant to extraction. JLA20 also stained the tips of microvilli on the apical surface of hair cells (Fig. 2c). These microvilli are shrinking and will be lost as the stereocilia bundle develops. Similarly, row 2 stereocilia lengths are also decreasing from P0-P9 as bundle architecture is refined[4], while row 1 stereocilia length remains constant. Stereocilia shortening and microvilli disassembly likely involve the actin severing proteins ADF and cofilin, which localize to the tips of row 2 stereocilia and apical surface microvilli. In addition, row 2 stereocilia and microvilli are both longer on hair cells lacking ADF and cofilin activity[9]. These observations suggest that the anchored G-actin detected by JLA20 could be a transient by-product of ADF/cofilin mediated actin disassembly in the tips of protrusions. It is unclear whether the liberation of G-actin during shortening contributes to concurrent tip filament formation or row 2 stereocilia widening.

Stereocilia morphogenesis is required for normal hearing and balance, and it is also a fascinating case study in how cells generate complex shapes. Here, we provided evidence of short actin filaments in developing stereocilia, both at the tips and along the widening shaft. We propose that these short filaments are intermediates that mature into the long actin filaments that have long been known to comprise the stereocilia actin core.

### Limitations of study

The primary limitation of this study is that the proposed widening mechanism partially relies on expression of exogenous actin and has not been validated through observation of endogenous actin incorporation. While our findings reveal a strong correlation of tip filaments to the occurrence of widening, the precise process by which actin adds to the periphery of core filaments and extends to the full length of stereocilia remains unclear.

## Methods

### Experimental model and subject details

Inbred C57BL/6 mice were used for all experiments. All animal procedures were approved by the Institutional Animal Care and Use Committee of Indiana University – Indianapolis (School of Science). Mice were housed in individually ventilated cages with free access to food and water. Pregnant females were provided with nesting materials and received breeder diet (#5015, Purina Lab Diet) instead of the standard diet (#5001, Purina Lab Diet). The animal facility operated on a 12:12-hour light/dark cycle. Ambient temperature and relative humidity were maintained at 22 ± 2 °C and 40–60%, respectively. The day of birth is referred to as postnatal day 0 (P0). Both male and female mice were used for all experiments.

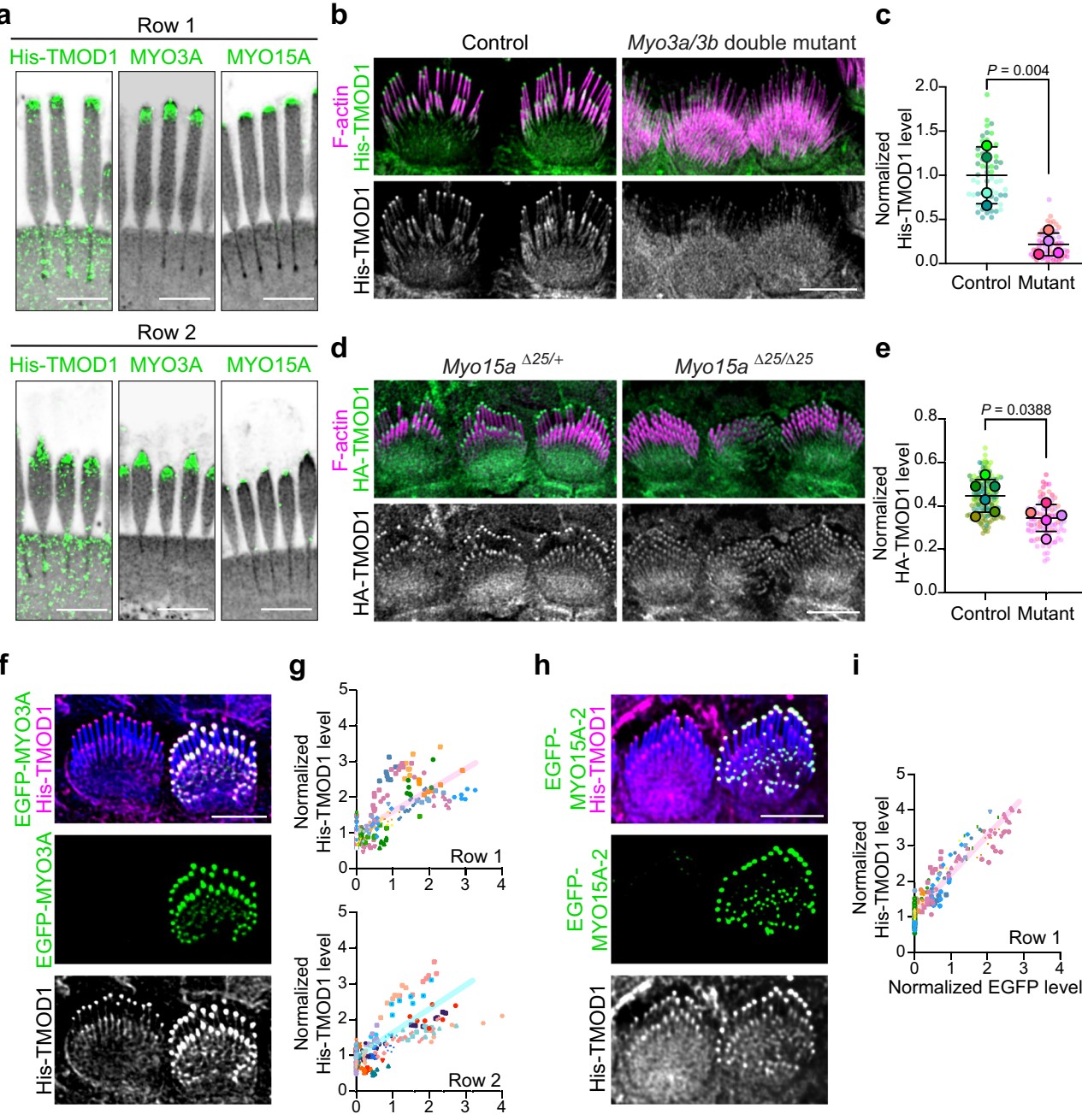

**Fig. 7 | Tip filaments depend on tip-localized myosins. a** Representative images of His-TMOD1 stained row 1 stereocilia and row 2 stereocilia, co-labeled after expansion microscopy with antibodies to endogenous MYO3A or MYO15A (green) from P5 mice. NHS-ester (grey) stained total protein. The experiments were repeated three times with similar results. **b** His-TMOD1 (green, grey) labeling in *Myo3* double mutant (*Myo3a*$^{\Delta14/\Delta14}$ *Myo3b*$^{\Delta12/\Delta12}$, noted in Supplementary Fig. 6a) and littermate control IHCs at P4. **c** Quantification of His-TMOD1 level from the tips of row 1 stereocilia in *Myo3a;Myo3b* G$_0$ mutants generated via i-GONAD (Supplementary Fig. 6a) and in littermate control IHCs, each at P4. Sample size (stereocilia, cells, cochleae/mice): control (60, 12, 4), mutant (60, 12, 4). **d** HA-TMOD1 (green, grey) labeling in *Myo15a* mutant (*Myo15a*$^{\Delta25/\Delta25}$) and littermate control (*Myo15a*$^{\Delta25/+}$) IHCs at P4. **e** Quantification of HA-TMOD1 level from the tips of row 1 stereocilia in *Myo15a* mutant and littermate control IHCs (P4). Sample size (stereocilia, cells, cochleae): control (125. 25, 6), mutant (85, 17, 5). In (**c**) and (**e**), normalized TMOD1 level from

individual stereocilia were plotted as small dots with the color corresponding to their cochleae, represented as larger open circles. Results were plotted with mean ± SD based on cochleae. *P* values for two-tailed unpaired *t* tests are indicated. *P* < 0.05 is considered statistically significant. **f** His-TMOD1 (magenta, grey) labeling of an IHC 18 h after transfection with EGFP-K50R-MYO3A (green) and a neighboring untransfected cell (P5). F-actin was stained by phalloidin (blue). **g** Graphs of His-TMOD1 and EGFP-K50R-MYO3A level at row 1 or row 2 stereocilia tips. Simple linear regression analysis was applied and R-squared values were 0.52 (row 1) and 0.51 (row 2). **h** His-TMOD1 (magenta, grey) labeling of an IHC 18 h after transfection with EGFP-MYO15A-2 (green) and a neighboring untransfected cell (P5). **i** Graph of His-TMOD1 and EGFP-MYO15A-2 level at row 1 stereocilia tips. Simple linear regression analysis was applied and R-squared value was 0.88. Sample size (stereocilia, cochleae) in (**g**) and (**i**): MYO3A (245, 10), MYO15A (238, 9). Scale bars represent 5 μm. Source data are provided as a Source Data file.

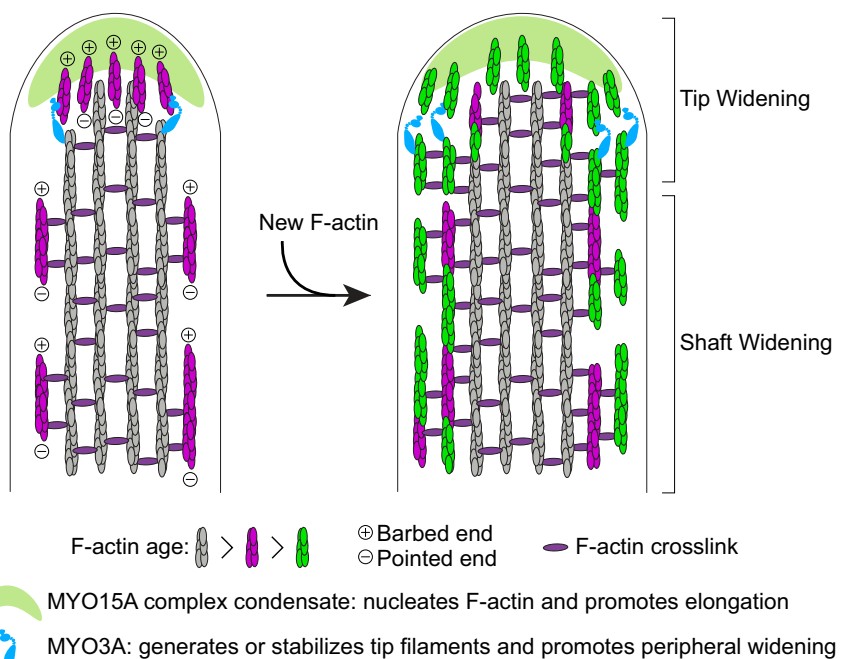

**Fig. 8 | Model for stereocilia widening by the addition of short actin filaments.** The drawing depicts a longitudinal section of a widening stereocilium and a hypothetical widening mechanism consistent with the data in this study. The grey filaments are long, stable core filaments bundled by F-actin crosslinks and new actin filaments are added to widen and stabilize the tip or the shaft. At the first timepoint, there are short actin filaments, shown in magenta, at the tip and along the periphery of the core. Tip widening occurs as these filaments form at the tip and are captured and stabilized by MYO3A. The source of tip filaments is unknown, but the MYO15A-dependent elongation complex is a favored candidate. Short filaments along the shaft, nucleated by an unknown mechanism, are extended by new actin to produce stable core filaments. Additional short filaments form along the periphery as widening continues.

The i-GONAD method was performed by following the published method[32,43]. Guide RNA (gRNA) targeting either *Myo15a* or *Myo3a* and *Myo3b* were made by mixing crRNA (listed below under materials) and tracrRNA (Integrated DNA Technologies, IDT) at 100 µM in duplex buffer (IDT), heating to 94 °C for 5 min, and cooling to room temperature. To assemble the ribonucleoprotein (RNP) complex, 2.25 µl gRNA and 0.75 µl recombinant *S. pyogenes* Cas9 endonuclease (IDT) were mixed in 4.5 µl OPTI-MEM medium (Gibco) and incubated at room temperature for 20 min. Fast Green (1 µl of a 1% filtered solution, Thermo Fisher Scientific) was added to aid visualization. Female mice with copulation plugs, estimated to be 14-16 h post mating, were anesthetized with isoflurane. The oviducts were exposed by an incision on the back of the mice. Each side of oviduct was injected with the RNP complex solution via a glass micropipette glass needle. The oviducts were electroporated with square wave pulses (8 pulses, duration 5 ms, interval 1 s, field intensity 50 V) delivered using tweezertrodes (BTX) connected to an ECM830 electroporator (BTX). The oviducts were repositioned, and the incision was closed with surgical staples. The resulting pups were genotyped by amplifying the targeted regions by polymerase chain reaction (PCR) using primers (*Myo15a*_sq_F/R, *Myo3a*_sq_F/R or *Myo3b*_sq_F/R) from genomic DNA isolated from tail snips. PCR products were purified using ReliaPrep DNA Clean-up and Concentration System (Promega, Cat. # A2892) and sequenced by amplicon sequencing service from Plasmidsaurus Inc. The mutations on each pup were identified by aligning the raw sequence reads.

## Materials

**Oligonucleotides.** Primers and guide RNAs were customized and purchased from IDT.

crRNA targeting *Myo15*: 5′-/AltR1/rCrUrArCrArArGrGrCrUrCrAr CrArCrGrGrUrGrG

rGrUrUrUrUrArGrArGrCrUrArUrGrCrU/AltR2/−3′

crRNA targeting *Myo3a*: 5′-/AltR1/rCrArCrCrGrCrCrUrGrArArArCrA rCrCrUrCuCrGrU rGrUrUrUrUrArGrArGrCrUrArUrGrCrU/AltR2/−3′

crRNA targeting *Myo3b*: 5′-/AltR1/rCrArUrCrUrCrGrGrUrGrGrArAr UrGrArArUrUrCrG

rGrUrUrUrUrArGrArGrCrUrArUrGrCrU/AltR2/−3′

*Myo15a*_sq_F/R: 5′-AGCAGGGGACCTATGACA-3′ / 5′-GAACCCCT-GAATAGCGTAACT-3′

*Myo3a*_sq_F/R: 5′-CAGGGCAAAGAAAGAGAATAAC-3′ / 5′-CATC-CAGACTACAGATACATGC-3′

*Myo3b*_sq_F/R: 5′-GTCAAGGGCCTTCTGAGG-3′ / 5′-CCTCACGTGT TGAAGCAATAG-3′

**Antibodies.** Primary antibodies for immunostaining were 1:200 rabbit anti-MYO15A-Pan (PB48), 1:100 rabbit anti-MYO3A (custom antibody from Genemed Synthesis Inc., raised against the C-terminal sequence, NPYDYRRLLRKTSQRQR, of mouse MYO3A), 1:100 mouse anti-TMOD1 (Thermo Fisher, Cat. # MA5-25612), 1:500 rabbit anti-GFP (Torrey Pines Biolabs, Cat. # TP401), 1:100 Alexa Fluor 488 mouse anti-β-actin (AC-15, 1.45 mg/ml, Novus Biologicals, Cat. # NB600-501), 1:50 monoclonal mouse anti-γ-actin antibody clone 1–37 IgG purified from ascites[44] and dye-conjugated (Thermo Fisher Antibody Labeling Kit Cat. #A20183), 1:200 Alexa Fluor 488/555 mouse anti-His (Genscript, Cat. # A01800 or # A01801) or 488 mouse anti-HA (Genscript, Cat. # A01806). Primary antibodies for immunoprobing unfixed, saponin permeabilized tissue were 100 nM mouse JLA20 from concentrated supernatant (198 µg/ml, Developmental Studies Hybridoma Bank, University of Iowa) and 100 nM Alexa Fluor 488 mouse anti-β-actin (AC-15). The secondary antibodies were 1:200 488/568/647 Alexa Fluor goat anti-rabbit or anti mouse IgG (Thermo Fisher).

**Plasmids for transfection.** EGFP-β-actin was a gift from Michael Davidson (Addgene plasmid # 56421)[45]. pEGFP-FSCN2 was cloned by shuttling a mouse Flag-*Fscn2* cDNA[46] to the pDest40 vector (Thermo Fisher). The pEGFP-C1-RPEL1-EGFP-3xNLS (Addgene plasmid #58469)[47] and then a stop codon was inserted after EGFP by PCR mutagenesis using the Q5 Site-Directed Mutagenesis Kit (NEB, E0554) to eliminate

expression of the nuclear localization signal (NLS). The pEGFP-C1-mTMOD1 construct[48] was donated by Dr. Velia M. Fowler (University of Delaware). The pEGFP-SH3BGRL2 was synthesized from GenScript based on the DNA sequence of SH3BGRL2 (SH3 domain binding glutamate rich protein like 2) from NCBI reference sequence: NM_172507.5. The expression of untagged actin was accomplished using pActin-IRES-EGFP, a dual expression construct with a single promotor that uses an internal ribosomal entry site (IRES) to drive EGFP expression. DNA encoding β-actin followed by the IRES was synthesized by GenScript and cloned into the N terminal of EGFP in the vector pcDNA3.1. Mutant β-actin plasmids, RFP-DVD-actin (DVD286,287,288AAA) and RFP-AP-actin (AP204,243EK), were previously made[27] and were gifted by Dr. Dmitri S. Kudryashov (The Ohio State University). EGFP-MYO3A, EGFP-K50R-MYO3A (kinase dead)[34,49], and EGFP-MYO15A-2 (short isoform)[10,29] were previously described.

### Cochlear culture and inner hair cell transfection by injectoporation

Auditory hair cells were transfected with plasmid DNA by the injectoporation technique[50]. Briefly, the sensory epithelium was dissected from C57BL/6 mice at postnatal day 3, 5, or 6 in Hank's Balanced Salt Solution (HBSS, Life Technologies, Cat. # 14025092), and the cochlear duct was opened by making an incision between Reissner's membrane and the stria vascularis. The tissue was explanted by adhering it to a plastic, tissue-culture treated dish (USA Scientific, Cat. # CC7672-3359) containing DMEM/F12 (Thermo Fisher Scientific, Cat. # 11039047) with 0.1 mg/ml penicillin. The culture was incubated at 37 °C with 5% $CO_2$ for 2 h before injectoporation was performed. For the injection step, a glass micropipette with a 2 µm tip diameter loaded with plasmid DNA (1-2 mg/ml in water) was oriented perpendicular to the IHC row. The tip of the micropipette was inserted into the space between two IHCs and pressure was supplied by a microinjector to inject plasmid into the tissue. An ECM 830 electroporator was used to deliver a series of three 15 ms 60 V square-wave electrical pulses at 1 s intervals to platinum wire electrodes that were 2 mm apart and positioned directly over the injection site. After the electroporation, the culture media was exchanged with Neurobasal-A medium (Thermo Fisher Scientific, Cat. # 12349015) supplemented with 2 mM L-glutamine (Thermo Fisher Scientific, Cat. # 25030081), 1x N-2 supplement (Thermo Fisher Scientific, Cat. # 17502048), 1.5 µg/ml D-glucose (Thermo Fisher Scientific, Cat. # 410955000), and 0.1 mg/ml penicillin.

### Live imaging on transfected inner hair cells

Cultured explants were transfected with EGFP-actin (2 mg/ml in water) by injectoporation. Cultures were incubated at 37 °C with 5% $CO_2$, then moved to a 37 °C microscope incubator for live-cell acquisition. Transfected cells were imaged at 2, 4, 18 h post injectoporation. Image stacks (0.13 µm intervals) were acquired in Airyscan mode with a Zeiss Apochromat 40x/1.0 NA water immersion objective on a Zeiss LSM 900 microscope. To avoid contamination, the cultured dishes were exchanged with fresh culture media after each live cell acquisition. Laser power for EGFP capture was 1% for time point 2 and 4 h and 0.1% for 18 h as EGFP-actin levels increased with time. Raw Airyscan images processed using Huygens Array Detector Deconvolution software within the Essential software package using automatic settings. Image stacks were processed in Imaris 10.0.1 for 3D construction and in Fiji for reslicing.

### Preparation of transfected inner hair cells for SIM imaging

Cultured explants were transfected with EGFP-actin or EGFP-FSCN2 (2 mg/ml in water) by injectoporation and fixed at 18 h post-transfection with 4% formaldehyde (Electron Microscopy Sciences, Cat. # 15710) in HBSS for 2 h and stained with Alexa Fluor 568 phalloidin (0.5 U/ml, Invitrogen, Cat. # A12380) in PBS with 0.1% Triton X-100 (Sigma, Cat. # X-100-100ML) at room temperature for 1 h. The tectorial membrane was removed and the tissue was mounted in Prolong Diamond (Thermo Fisher Scientific, Cat. # P36961). Calibration slides for channel alignment were prepared using tissue stained with Alexa Fluor 488 and 568 phalloidin (0.5 U/ml, Invitrogen) following the same procedures from above.

### Ultrastructural expansion microscopy (U-ExM) and immunolabeling

Ultrastructural expansion microscopy (U-ExM) was performed based on protocols adapted from previously described methods[51,52]. Cochlear tissue was fixed at room temperature for 30 min. After fixation, tissue was washed with PBS and transferred to 1.4% formaldehyde/2% acrylamide (FA/AA) in PBS at 37 °C for overnight incubation. The FA/AA solution was removed on the second day followed by 3 times wash of PBS. The tissue was then incubated in monomer solution (19% w/w sodium acrylate (Pfaltz & Bauer, Cat. # S03880), 10% v/v acrylamide (Sigma, Cat. # A4058), 0.1% v/v N, N′-methylenebisacrylamide (Sigma, Cat. # M1533) in PBS) for 90 min at room temperature. After incubation, the cochlear tissue was transferred to a petri dish lid. Excess monomer solution was removed from the surrounding area of the tissue. Subsequently, 2.5 µl of 10% N,N,N′,N′-Tetramethylethylenediamine (TEMED) and 10% ammonium persulphate solution were added to 45 µl monomer solution, which was mixed thoroughly, and 30 µl was quickly pipetted onto the dish before being mounted under a 12 mm round coverslip. After polymerization, the resulting hydrogel was then incubated at 37 °C humidified chamber for 1 h before adding room temperature denaturation buffer (200 mM sodium dodecyl sulfate (SDS), 200 mM NaCl, 50 mM Tris, pH 9) for 15 min. The gel was then detached from the coverslip, transferred to a new denaturation buffer, and denatured at 95 °C for 1 h. The denatured gel was transferred in a 150 mm petri dish filled with 20 ml MilliQ water and placed on a 50-rpm shaker for 30 min. The gel was further expanded by replacing water two more times every 30 min. The expanded gel was shrunk by washing with PBS for 15 min. The shrunken gel was then incubated with 0.02 mg/ml Alexa Fluor 546 succinimidyl ester (NHS-ester, Thermo Fisher, Cat. # A20102) for 1 h before washing 3 times with PBS. The gel was incubated with blocking buffer (2.5% BSA in PBS, 0.5% Triton-X-100) for 1 h blocking at room temperature on the shaker and washed with PBS 3 times before antibody staining. Primary antibodies were prepared in the staining buffer (1% BSA in PBS, 0.2% Triton-X-100) and incubated with the gel overnight at room temperature on the shaker. The gel was washed with PBS 3 times before being incubated with secondary antibodies in PBS for 3 h at room temperature on the shaker. Following the incubation, the stained gel was washed with PBS and then transferred to the 10 cm petri dish filled with water, undergoing a second round of expansion during 3 subsequent water washes. The gels were trimmed, and mounted in a glass-bottom dish under a glass coverslip. To estimate sample expansion parameters, tissue and gels were measured before and after expansion. Tissue expanded by a factor of 3.8x compared to 3.9x for gels without tissue.

EGFP was detected by rabbit-anti-GFP primary antibody followed by the incubation of the secondary antibody goat anti-rabbit Alexa Fluor 488 or 568. His-TMOD1 was incubated with unfixed tissue at P5. After probing, the tissue was fixed and processed as described above. Anti-TMOD1, anti-MYO15A-Pan, or anti-MYO3A antibodies were incubated with gels for 24 h at room temperature on the shaker, washed, and incubated with the secondary goat anti-rabbit or anti-mouse Alexa Fluor 488 antibody for 3 h. The whole expanded tissue was stained by Alexa Fluor 546 conjugated NHS-ester to label total protein.

## Expression and purification of His-TMOD1, HA-TMOD1, and His-CAPZ

The expression construct pReceiver-B01-mTmod1 and purified His-TMOD1 were gifted by Dr. Alla Kostyukova (Washington State). HA-Tmod1 has an N-terminal 6xHis tag followed by a fusion of the "spaghetti monster (sm)" HA tag[53] with mTmod1. This construct was generated by amplifying the smHA sequence from a plasmid pCAG-smFP-HA (Addgene #59759) with forward primer (5'-ccatcac-catcattcgaagg*aaggt*ACCATGTACCCTTATGATGTGC-3') and reverse primer (5'- gtctgtaggacatggtaccg*cctgcccc*AGCGTAGTCCGGGACATC-3'). The amplified product was then assembled with EcoRI-digested pReceiver-B01-mTmod1 using the NEBuilder HiFi DNA Assembly Kit (New England Biolabs, Cat. # E5520S). The entire plasmid was sequenced using nanopore sequencing (sequencing performed by Plasmidsaurus). Competent *E. coli* Rosetta (DE3) pLysS cells were transformed with the His-Tmod1 or HA-Tmod1 plasmid. Successful transformants were selected by incubating bacteria at 37 °C on LB agar plates containing ampicillin. A single colony was used to grow a 1-liter culture in LB-ampicillin medium, which at $OD_{620nm}$ 0.6 was induced to express protein with 0.4 mM IPTG and further incubated for 5 h at 37 °C. The cells were harvested by centrifugation and resuspended in 20 ml lysis buffer (50 mM Na-phosphate (pH 6.8), 300 mM NaCl, 5 mM 2-mercaptoethanol (BME), 1 mM phenylmethylsulfonyl fluoride (PMSF), 1 tablet cOmplete protease inhibitor, 20 mM imidazole, 10% glycerol). The resuspended cells were sonicated and then centrifuged to remove cellular debris. The supernatant was loaded onto a Ni-NTA column which was pre-equilibrated with the wash buffer (50 mM Na-phosphate (pH 6.8), 300 mM NaCl, 5 mM BME, 1 mM PMSF, 20 mM imidazole, 10% glycerol). After washing the column with 50 ml of wash buffer, the His-TMOD1 protein was then eluted with 70 ml elution buffer with a gradient of 20-250 mM imidazole (50 mM Na-phosphate (pH 6.8), 300 mM NaCl, 1 mM BME, 1 mM PMSF, 10% glycerol, 250 mM imidazole for the maximum concentration). The elution was collected in 90 fractions. The purity of the His-TMOD1 or HA-TMOD1 fractions was assessed by SDS-PAGE. Fractions with over 95% purity were pooled together and dialyzed overnight in the storage buffer (20 mM Tris HCl (pH 8.0), 1 mM EDTA, 1 mM dithiothreitol (DTT), 10% glycerol). The concentration of the dialyzed His-TMOD1 or HA-TMOD1 was measured, and the aliquots were stored at −80 °C. The His-CAPZ was previously made[9] and aliquots used in this study were thawed from stocks stored at −80 °C.

## Purified protein probing assay

Cochleae were dissected in HBSS and the lateral wall was removed prior to permeabilization in cytoskeleton buffer (20 mM HEPES (pH 7.5), 138 mM KCl, 4 mM $MgCl_2$, 3 mM EGTA, 1% bovine serum albumin) with 0.05% saponin. Purified proteins were included at the follow concentrations: 714 nM His-TMOD1 (6XHis-tagged mouse tropomodulin-1) to label F-actin pointed-ends, 3.6 µM Alexa Fluor 488 conjugated DNase I (deoxyribonuclease I, Thermo Fisher, Cat. # D12371), or 400 nM His-CAPZ (mouse CAPZB and 6XHis-tagged CAPZA1) to label barbed ends of actin filaments. Samples were incubated in this solution for 5 min at room temperature before being washed by cytoskeleton buffer without saponin. Samples were then fixed with 4% formaldehyde in HBSS for 2 h at room temperature. Samples were rinsed with PBS and the tectorial membrane and Reissner's membrane were removed from the fixed tissues. Samples then were incubated overnight at 4 °C with 5 ng/ml Alexa Fluor 488 conjugated anti-His antibody and 1 U/ml Alexa Fluor 568 phalloidin in PBS with 0.01% Triton X-100. Tissues were rinsed by PBS 3 times and mounted in Prolong Diamond for imaging. To visualize His-TMOD1 and His-CAPZ in EGFP-Myosin transfected cells, 5 ng/ml Alexa Fluor 647 conjugated anti-His antibody was used instead. Samples labeled with Alexa Fluor 488 conjugated DNaseI were incubated with 1 U/ml Alexa Fluor 568 phalloidin in PBS with 0.01% Triton X-100 at room temperature for 1 h

before mounting. Apical IHCs were defined as being at a cochlear location approximately 30% of the distance from the apical end.

## Pre-fix antibody labeling

Freshly dissected sensory epithelia were permeabilized in cytoskeleton buffer with 0.05% saponin, subsequently mixed with 100 nM JLA20 or FITC-AC15. Samples were incubated in this solution for 5 min at room temperature before being washed by cytoskeleton buffer without saponin. Samples were then fixed with 4% formaldehyde in HBSS for 2 h at room temperature. Samples were rinsed with PBS and the tectorial membrane and Reissner's membrane were removed from the fixed tissues. Samples were then incubated overnight at 4 °C with 1:200 secondary antibody goat anti-mouse Alexa Fluor 488 (to detect JLA20) and 1 U/ml Alexa Fluor 568 phalloidin in PBS with 0.01% Triton X-100. Tissues were rinsed with PBS 3 times and mounted in Prolong Diamond for imaging.

## Permeabilization of transfected cells

Explants were transferred from culture dishes to HBSS 18 h post-transfection with RPEL1-EGFP or EGFP-TMOD1, then rinsed in cytoskeleton buffer with 0.05% saponin and immediately fixed with 4% formaldehyde in HBSS for 2 h at room temperature.

## Latrunculin A treatment assay

The cultured explants were categorized into 3 groups: DMSO, LatA (Latrunculin A, Sigma, Cat. # 428026), and washout. After 1-hour pre-culture, solutions from the DMSO group were exchanged with Neurobasal-A medium containing DMSO (1:1000) at a final concentration of 0.1%. The solutions from both LatA and washout group were exchanged with Neurobasal-A medium containing LatA (1:1000) with a final concentration of 1 µM. After a 1-hour incubation at 37 °C and 5% $CO_2$, the DMSO and LatA groups were then harvested and transferred to the cytoskeleton buffer for His-TMOD1 probing assay described above. The explants from the washout group were washed with warm Neurobasal-A medium 6 times to decrease the LatA concentration to be less than 0.01 µM. The dishes were further cultured for 4 h before performing His-TMOD1 probing assay.

## High salt stripping assay

Cochleae were dissected from C57BL/6 mice at postnatal day 5 and lateral walls were removed to expose the sensory epithelia. One cochlea was transferred to a high-salt buffer (20 mM HEPES (pH 7.5), 138 mM KCl, 4 mM $MgCl_2$, 3 mM EGTA, 1% bovine serum albumin, 500 mM NaCl and 0.2% saponin for 2 min incubation at room temperature, while the other cochlea was transferred to the same buffer without NaCl and 0.05% saponin for 2-minute incubation as a control. After high-salt extraction, tissues were washed twice with cytoskeleton buffer containing 0.05% saponin before being probed for 5 min at room temperature with His-TMOD1 or His-CAPZ. Samples were then washed with cytoskeleton buffer without saponin and fixed with 4% formaldehyde at room temperature for 2 h. The subsequent steps after fixation were as described above in the purified protein probing assay. In the high salt experiment, the anti-His antibody was Alexa Fluor 555 conjugated and the phalloidin was Alexa Fluor 488.

## Orthovanadate treatment assay

Dissected P5 organs of Corti were transferred to cytoskeleton buffer (20 mM HEPES (pH 7.5), 138 mM KCl, 4 mM $MgCl_2$, 3 mM EGTA, 1% bovine serum albumin) and 0.05% saponin with or without 4 mM sodium orthovanadate[54] (New England Biolabs, Cat. # P0758S). After a 1-minute incubation, His-TMOD1 (714 nM) was added to the solution and incubated at room temperature for 5 min. Samples were then washed with cytoskeleton buffer without saponin and fixed with 4% formaldehyde at room temperature for 2 h. The subsequent steps after fixation were as described above in the purified protein probing assay.

## Fluorescence microscopy

Slides in Figs. 3d, 4, 5, 7f, h, Supplementary Figs. 2, 3, 4a, 6c, 6f, 7a were imaged with a Leica Plan Apo 63x/1.40 NA oil immersion objective on Leica SP8 inverted confocal microscope operating in resonant scanning mode (Leica Microsystems, RRID:SCR_018169). Images were captured using Leica Application Suite X (Leica Microsystem, RRID:SCR_013673) and deconvolved using Leica LIGHTNING deconvolution with the default settings. Slides in Figs. 2, 3a, 6, 7b, d, Supplementary Figs. 1, 4b, 4c 5a, and 6e were imaged with a Zeiss Plan-Apochromat 63x/1.4 NA oil immersion objective on a Zeiss LSM 900 microscope with an Airyscan detector. Raw Airyscan images were processed in Zen software using default settings. U-ExM samples were imaged with a Leica HC FLUOTAR L 25x/0.95 NA water immersion objective on the Leica SP8. Slides in Supplementary Fig. 6d were imaged using a Nikon Apochromat TIRF 100x/1.49 NA oil immersion objective on a Nikon Ti2-E with a spinning disk confocal scan head (CSU-X1, Yokogawa) and captured on a sCMOS camera (Prime95B, Teledyne Photometrics). Structured illumination (SIM) images were acquired at 26 to 30 °C with a 63x/1.4 NA oil immersion lens on a Zeiss (Oberkochen, Germany) lattice-based Elyra 7 microscope with dual PCO.edge 4.2 sCMOS cameras for detection. Illumination grid selection and z-spacing was guided by the software and kept consistent across images. Post-acquisition processing was performed with software-recommended standard filtering for the 488-nm channel, without baseline subtraction and with "scale to raw" checked. Contrast was manually adjusted to retain both dim and bright structures due to the high dynamic range of the phalloidin signal. Verification of channel alignment was carried out as previously described[4]. ImageJ (NIH, RRID:SCR_002285) was used to adjust colors and display values for the images presented in figures.

## Image analysis

3D reconstructions in Fig. 1 were made with Imaris software. All images for fluorescent quantification were analyzed using ImageJ.

**Line scan profiling on stereocilia.** Line scans were drawn from the top of stereocilia down the center of stereocilia. In Figs. 2b, d, 3b, c, the peak level in green channel was set as 0 on x axis and the fluorescence intensity was normalized to the maximal fluorescence intensity of row 1. In Fig. 3g, the stereocilia tip was defined as being the point where phalloidin intensity reached the mean value of fluorescence in the tip region. The fluorescence intensity was normalized to the average fluorescence intensity of line scan values in each channel. In Fig. 6c, the peak level in RFP channel was set as 0 on axis and the fluorescence intensity was normalized to the maximal intensity in each channel.

**Tip or shaft signal analysis.** The signals at stereocilia tips were collected by drawing a 10×10 pixels circle (0.4 μm in diameter). The shaft signals in stereocilia during postnatal development were collected by drawing a line (around 1 μm) along the shaft. Mean value of either circle or line was measured on the maximum intensity projection. In Fig. 2e, the fluorescence intensity of JLA20 was normalized to the average intensity of the cell border by cochlea. In Fig. 4b, His-TMOD1 signals were normalized to the average fluorescence intensity of row 1 stereocilia from DMSO-treated samples. In Fig. 4d–f, His-TMOD1 signals were normalized to the average fluorescence intensity of row 1 stereocilia at P3. In Fig. 5, the shaft signals were collected by drawing a 10×10 pixel circle (0.45 μm in diameter) at a point of 0.45 μm below stereocilia tip. The average shaft intensity of His or EGFP signals was measured and normalized to the average fluorescence intensity of the corresponding His labeling in all cells or EGFP labeling in transfected cells, respectively. The shaft level of EGFP or His labeling in each stereocilium was determined by multiplying the average shaft intensity by the width of the stereocilium. In Fig. 7c, His-TMOD1 signals were normalized to the average fluorescence intensity of row 1 stereocilia in control. In Fig. 7e, HA-TMOD1 levels were normalized to the average fluorescence intensity of cell junctions in each image acquisition. In Fig. 7g, i, the circle size for collecting tip signals was increased to 14×14 pixels (0.6 μm in diameter). EGFP signals were normalized to the average EGFP levels at stereocilia tips of the same row in all transfected cells. His-TMOD1 signals were normalized to the average fluorescence intensity of stereocilia tips in control cells.

**Stereocilia width measurement.** The stereocilia width was defined by measuring full width at half maximum (FWHM) at a position 0.45 μm below stereocilia tip. The FWHM was measured using a nonlinear curve fitting with the Gaussian function in OriginLab 2022. The measurements for samples imaged by expansion microscopy in Supplementary Fig. 7 were performed as follows: The "Fire" LUT function was applied to the maximum intensity projection. The image display range was set from 0 to 40% of the maximum displayed value. A line was drawn perpendicular to the stereocilia, across pixels with saturated intensity (at least 40% of the maximum value). The line length was measured at a position 1.41 μm below stereocilia tip.

## Statistics and reproducibility

Analysis was determined by the availability of quantifiable cochleae; no statistical method was used to predetermine sample size. Only cochlear samples in which stereocilia were orientated parallel to the coverslip such that the full length was captured within a z-stack that was less than 0.5 microns thick were chosen for analysis. Maximum intensity projections of selected stacks covering the entire stereocilia length were created for analysis on individual cell. Each experiment was conducted at least two times to validate replicability. The data were collected from at least 3 wildtype cochleae and 3 mutant littermates per measurement, at least 3 cells per cochlea, and 3-7 stereocilia per cell. Statistical analysis was performed with GraphPad Prism 10. SuperPlots[55] were applied to present the data with column table format from GraphPad. Smaller circles represent stereocilia or cells and larger circles represent mice or cochleae in all graphs. Each color corresponds to a replicate unit (mouse or cochlea as indicated). For plots in Figs. 5b–d, 7g, and 7I, each symbol corresponds to a stereocilium. Stereocilia from the same cell are represented by identical color and shape. A two-tailed Student's t test was used for pairwise comparisons. A two-way two-sided ANOVA was used to determine significant differences in Fig. 3c. Data are presented as mean ± standard deviation (SD). $P < 0.05$ is considered statistically significant.

## Reporting summary

Further information on research design is available in the Nature Portfolio Reporting Summary linked to this article.

## Data availability

Custom antibodies, DNA constructs, and mouse mutants that support this study are available upon request to the corresponding author. Raw image data was deposited in Figshare [https://doi.org/10.6084/m9.figshare.28680503]. Source data are provided with this paper.

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

## Acknowledgements

This work was supported by Pennsylvania Lions Hearing Research Grant (C.M.Y), R01DC011034 (P.G.B.G), R01DC002368 (P.G.B.G), R01DC018827 (J.E.B), and R01DC015495 (B.J.P).

## Author contributions

X.L. and B.J.P. developed the concept, designed the study, and wrote the paper, which all authors edited. X.L. and B.J.P. produced the figures. J.A.C., M.C., C.M.Y., J.F.K., P.G.B.-G., J.E.B and B.J.P. contributed reagents. C.-Y.T. performed the i-GONAD procedure. X.L., C.-Y.T., J.F.K., and G.B., collected data, and all authors contributed to data analysis.

## Competing interests

The authors declare no competing interests.
