## [Transparent Peer Review file · Nature Communications]

Myosin-dependent short actin filaments contribute to peripheral widening in developing stereocilia

Corresponding Author: Dr Benjamin Perrin

Version 0:

Reviewer comments:

Reviewer #1

(Remarks to the Author)

Understanding the mechanism of stereocilia (SC) widening during cochlear inner hair cell bundle formation is a longstanding problem in the cytoskeleton field. Using a combination high resolution imaging, actin probes and mutants, Liao et al. show that widening during stage III of SC maturation (P5 - P6) likely results from the production of new actin filaments that then surround, but are not a part of, the actin core. These filaments are largely at the tip of SC, but are also seen along the length of the SC. Consistent with a role in SC widening, tip filaments are transient, decreasing significantly as SC widening is completed (P8, P9). Inhibition of actin polymerization blocks filament widening and ectopic expression of G-actin increases it, clearly establishing the essential role of actin polymerization in this process. SC from mice mutant for myosins associated with SC growth and widening (Myo15A and Myo3A/B) lack tip filaments, suggesting the motors have a role in the production or stabilization. Together, these results provide a clearer view of events that occur during SC tip widening in inner hair cells.

The work addresses a fundamentally important and interesting problem - how do the staged events of SC maturation occur? The results are clear and the data presented are convincing. It will not come as a surprise to anyone that actin polymerization is essential for widening, but the proposed appearance of new filaments, especially at the SC tip, is a significant observation and suggests a finely tuned mechanism for the staged and local production of these filaments. It will be interesting to learn about how this might happen. Overall, the work is an elegant example of how tight, spatially localized control of actin filament production during the assembly of cytoskeletal structures can occur.

1) It should be noted that while the observed staining for actin barbed and pointed ends is quite convincing and it is logical to conclude that these are indicators of new filaments at the tip, there is no direct evidence that this is the case. One could argue that pointed ends seen in the tip region and along the shaft are the result of filament severing that increases actin dynamics. While this seems unlikely, can the authors rule out the possibility that what they are seeing is remodeling of the actin core that occurs by severing of existing core actin filaments during the widening stage?

2) Is there any EM evidence for the existence of short filaments in the tip region during widening (most of the work seems to be on later stage SC)?

3) The need for Myo15A for filament production is not unexpected given its known role in delivering the elongation complex to the SC tip and Myo3A has also been shown to be important for widening and regulating overall SC length. The absence of tip filaments in the null mutant suggests a role for these myosins in their production but it is unclear whether they have a direct role or if they indirectly contribute. If the myosins are generating force to break the actin filaments, generating new ends, then is this process thought only happen at tip given the localization of the myosins?

4) How do the authors envision the additional filaments are added to the lower part of the SC shaft? Do they think that new filaments are nucleated in various regions along the SC? Is anything known about the expression of actin regulators during stage III that might provide clues about how the production of the proposed filaments might occur?

5) It would be helpful if the authors could provide an illustration of how they now think the widening process occurs given the findings here.

Reviewer #2

(Remarks to the Author)

Liao et al. describe a detailed procedure for peripheral widening of stereocilia. The authors used diverse imaging techniques, including SIM and expansion microscopy, to visualize nanoscale details of stereocilia and their changes. They identified a role for short actin filaments during the morphogenesis of stereocilia. Their work is very interesting and detailed and provides solid support for their finding. However, I have some questions about the imaging techniques used in this work, particularly expansion microscopy.

1. The use of expansion microscopy in this work is limited: Figure 1c, Figure 7a, and Supplementary Figure 6b. The authors stated that they employed a protocol that is used for expanding parasites. In expansion microscopy, the expansion protocol has to be carefully optimized to ensure deformation-free expansion, especially tissue. In this work, the authors did not use expansion microscopy to measure the size of ultrasmall structures. Thus, expansion uniformity was not critical. However, providing an expansion factor would be a good indication of whether the expansion was truly uniform or not. If the expansion factor were smaller than the expansion factor of the gels without tissue, it would indicate that tissue expansion was not uniform. Therefore, can the authors please clarify the expansion factor in the expansion microscopy experiments and compare it with that of the black hydrogel, which does not contain any tissue?
2. Figure 1b shows a top-down view of a clear ring structure, which was imaged using an Airyscan microscope. Such ring structures would be more clearly shown using expansion microscopy. However, such structures are not clearly visible in the 3D view of the expansion microscopy result shown in Figure 1c. Are such ring structures apparent in the expansion microscopy images?
3. Related to the above question, other experiments, such as the one shown in Figure 3f, could have benefited from the application of expansion microscopy. Why was expansion microscopy not used in these experiments requiring highly precise measurements?
4. In this study, the authors used Alexa 546 NHS ester to label all proteins in expansion microscopy. The use of NHS ester is widespread in expansion microscopy. However, none of the results included in this study show an Alexa 546 NHS ester channel. Is there any reason the Alexa 546 NHS ester channel is not shown?
5. In the Methods section, the concentration of bisacrylamide is stated to be 2%. I believe this is incorrect, as the typical stock concentration of bisacrylamide is 2%. Please check the concentration and amend if not correct.
6. I am unable to find a data availability statement and associated data in this manuscript. It would be useful to see the raw data. Please include a data availability statement.

Reviewer #3

(Remarks to the Author)

Remarks to the Author:

The mechanosensitive function of hair cells relies on stereocilia. However, the mechanisms underlying the lengthening and widening of stereocilia in inner ear hair cells remain an unresolved question in cell biology. In this paper, the authors focus on the widening of stereocilia and demonstrate that newly synthesized actin first incorporates at the stereocilia tip, followed by incorporation along the shaft, thereby promoting stereocilia widening. The authors also identified a previously undetected population of short actin filaments and proposed that these filaments act as intermediates in the assembly of new stereocilia core filaments. However, the reviewer has the following concerns regarding this work.

1. Line 12: The research does not involve the vestibular system, and it is recommended that the term "vestibular system" be removed.
2. Line 80: In reference 5, the authors studied the incorporation of β -actin-GFP into stereocilia in organotypic cultures of hair cells from rat and mouse inner ear sensory epithelia, observing the incorporation and flux of actin, which results in the distribution of actin-GFP along the entire stereocilia. Given these findings, could the dynamic expression of EGFP-actin in the IHCs in the present study also be attributed to actin turnover?
3. How can the possibility be excluded that the observed actin incorporation results from anomalous actin overexpression?
4. Fig. 1b: Does the pattern of EGFP-actin incorporation into the stereociliary bundle depend on the age of the mice?
5. Line 83: What was the rationale for selecting P6 inner hair cells for transfection?
6. Lines 84-88: In conjunction with reference 5, hair cell cultures were transfected with β -actin-GFP. Forty-eight hours after transfection, the static cilia bundles were fully labeled. In contrast, in the present study, inner hair cells showed labeling at the tip and axis of the ciliary bundles only 18 hours post-transfection. Could this difference be due to the timing of transfection?
7. Lines 106-107: According to what principle does the conclusion that "probe or G-actin is easily diffusible" arise?
8. Lines 106-110: When cells were extracted with saponin, the RPEL1-EGFP signal was drastically reduced to nearly

undetectable levels. Under these conditions, JLA20 staining was enriched at the tips of row 2 stereocilia. What could explain this observation?

9.Fig. 2a: How long after transfection of RPEL1-EGFP were the hair cells sampled for staining?

10.Lines 118-119: Saponin can deplete the soluble G-actin pool in cells. Why, after saponin extraction, did the tip enrichment of TMOD1 signals become more pronounced, with a greater reduction in shaft signals? Additionally, how does this affect the signaling of F-actin?

11.Lines 124-128: Are these tip filaments present exclusively at the tips of developing stereocilia? Additionally, are these filaments also found in the stereocilia of adult hair cells?

12.Lines 147-148: How much time after the drug washout did the level of the pointed end recover significantly?

13.Fig. 4a, 4b: LatA, a drug that sequesters G-actin and depolymerizes dynamic F-actin structures, had no effect on the core actin filaments. What could explain this lack of effect?

14.Fig. 5a: This panel illustrates that overexpression of actin leads to stereocilia widening and an increase in the number of tip filaments. However, there is no direct evidence establishing a link between the tip filaments and the observed stereocilia widening.

15.Fig. 6a: Compared with IHC transfected with EGFP-actin alone, IHC transfected with both EGFP-actin and RFP-AP-actin exhibit wider stereocilia bundles with greater incorporation of EGFP-actin. Could you provide an explanation for this observation?

16.Line 185: In addition to MYO3A/B, MYO15A, and many other proteins such as ESPIN, EPS8, and GPSM2 that regulate stereocilia elongation and widening, why were MYO3A/B and MYO15A specifically selected for regulatory analysis?

17.Lines 54, 55, 243-245: A prolonged transfection time may be necessary to observe the pattern of newly incorporated actin, which would help support the conclusion.

18.There is no direct evidence to suggest that the reduction of tip filaments in the stereociliary bundle influences its elongation and widening.

19.As a general point, there are several datasets where the sample size (n) is not provided. Could the authors please ensure that the sample size (n) is included for all datasets, along with a clear explanation of what the number represents?

20.Fig. 1b suggests that incorporation patterns vary at different time points following plasmid transfection. Please specify the time points at which samples were collected in all relevant results.

Version 1:

Reviewer comments:

Reviewer #2

(Remarks to the Author)

My comments have been fully addressed.

Reviewer #3

(Remarks to the Author)

The author has already solved all my problems

Point-by-point response to the reviewers' comments

We thank the reviewers for their insightful and constructive comments on our manuscript. In response, we made several substantive changes, including the addition of new data and discussion points. Point-by-point responses are below in blue font. Changes to the attached manuscript are highlighted in yellow.

Reviewer #1 (Remarks to the Author):

Understanding the mechanism of stereocilia (SC) widening during cochlear inner hair cell bundle formation is a longstanding problem in the cytoskeleton field. Using a combination high resolution imaging, actin probes and mutants, Liao et al. show that widening during stage III of SC maturation (P5 - P6) likely results from the production of new actin filaments that then surround, but are not a part of, the actin core. These filaments are largely at the tip of SC, but are also seen along the length of the SC. Consistent with a role in SC widening, tip filaments are transient, decreasing significantly as SC widening is completed (P8, P9). Inhibition of actin polymerization blocks filament widening and ectopic expression of G-actin increases it, clearly establishing the essential role of actin polymerization in this process. SC from mice mutant for myosins associated with SC growth and widening (Myo15A and Myo3A/B) lack tip filaments, suggesting the motors have a role in the production or stabilization. Together, these results provide a clearer view of events that occur during SC tip widening in inner hair cells.

The work addresses a fundamentally important and interesting problem - how do the staged events of SC maturation occur? The results are clear and the data presented are convincing. It will not come as a surprise to anyone that actin polymerization is essential for widening, but the proposed appearance of new filaments, especially at the SC tip, is a significant observation and suggests a finely tuned mechanism for the staged and local production of these filaments. It will be interesting to learn about how this might happen. Overall, the work is an elegant example of how tight, spatially localized control of actin filament production during the assembly of cytoskeletal structures can occur.

1) It should be noted that while the observed staining for actin barbed and pointed ends is quite convincing and it is logical to conclude that these are indicators of new filaments at the tip, there is no direct evidence that this is the case. One could argue that pointed ends seen in the tip region and along the shaft are the result of filament severing that increases actin dynamics. While this seems unlikely, can the authors rule out the possibility that what they are seeing is remodeling of the actin core that occurs by severing of existing core actin filaments during the widening stage?

Thank you for this insightful comment. We agree that severing is a potential mechanism to generate pointed ends. This mechanism may be particularly relevant for IHC row 2 stereocilia because ADF and cofilin localize to those tips and these stereocilia shorten slightly during Stage III development (McGrath et al. *Current Biology*. 2021. PMID: 33400922 and Krey et al. *PLoS Biology*. 2023. PMID: 37011103). The following sentence was added to the discussion to suggest that severing could contribute to pointed end generation: (Lines 256-257) “Alternatively, existing core filaments could be severed by ADF or cofilin, both of which localize to row 2 stereocilia tips and regulate stereocilia growth.”

We think severing is less likely to generate tip filaments at row 1 tips since ADF/Cofilin is not detected there and tip filaments are generally more abundant than at row 2 tips. In our previous study (McGrath 2021, above), we found that IHC stereocilia lacking ADF and cofilin were of similar

width to control stereocilia, arguing that severing along the shaft is not contributing much to actin core remodeling in this cell type.

2) *Is there any EM evidence for the existence of short filaments in the tip region during widening (most of the work seems to be on later stage SC)?*

We have been unable to find or generate EM data to directly visualize tip filaments. The most recent cryoEM structures reconstruct the core actin filaments near the tip, but do not reconstruct the tip density (Elferich *et al. Elife*. 2021. PMID: 34964715). Classic EM micrographs from Tilney, Hirokawa, and colleagues have very few images of inner hair cell stereocilia from mammals, and none at the developmental ages when these structures are widening and actin filaments are discernable.

3) *The need for Myo15A for filament production is not unexpected given its known role in delivering the elongation complex to the SC tip and Myo3A has also been shown to be important for widening and regulating overall SC length. The absence of tip filaments in the null mutant suggests a role for these myosins in their production but it is unclear whether they have a direct role or if they indirectly contribute. If the myosins are generating force to break the actin filaments, generating new ends, then is this process thought only happen at tip given the localization of the myosins?*

We are intensely curious about the mechanisms by which MYO3A and MYO15A contribute to tip filament levels. Certainly, these proteins could be acting indirectly by delivering other factors that regulate tip filament levels. Even if they are acting directly, several alternative mechanisms are possible, such as binding and stabilizing tip filaments, nucleating, or breaking existing filaments. So, there is a lot of work to do in subsequent studies to sort out how myosins are regulating tip filaments.

Regarding filament breakage, we think it would be limited to the tip where there is the greatest concentration of motors, and presumably force. In addition, long actin filaments along the shaft are probably crosslinked to adjacent filaments and would thus be more resistant to breakage.

4) *How do the authors envision the additional filaments are added to the lower part of the SC shaft? Do they think that new filaments are nucleated in various regions along the SC? Is anything known about the expression of actin regulators during stage III that might provide clues about how the production of the proposed filaments might occur?*

These are excellent questions that we hope to address in future studies, but we are happy to speculate here. One possibility is that short filaments generated at tips move to the shaft and serve as seeds for widening. However, attempts to visualize filament movement using photoactivated probes and Airyscan microscopy yielded inconclusive results. Our sense is that we will need brighter probes that could allow for single molecule imaging, and likely a different imaging modality. We have explored the role of widely known actin nucleators (ARP2/3 and formins) but don't have reliable observations to suggest these have a role in nucleating short filaments in stereocilia at tips or along the shaft. If short filaments are nucleated at various regions along the shaft, which seems like a plausible scenario, then nucleation would depend on shaft localized nucleation factors. Unfortunately, as of now, we have not identified such a factor.

5) *It would be helpful if the authors could provide an illustration of how they now think the widening process occurs given the findings here.*

Thank you for this suggestion. We have added an illustration depicting a proposed widening process as Figure 8 (included below).

Fig. 8: Model for stereocilia widening by the addition of short actin filaments

The drawing depicts a longitudinal section of a widening stereocilia and a hypothetical widening mechanism consistent with the data in this study. The grey filaments are long, stable core filaments bundled by F-actin crosslinks and new actin filaments are added to widen and stabilize the tip or the shaft. At the first timepoint, there are short actin filaments, shown in magenta, at the tip and along the periphery of the core. Tip widening occurs as these filaments form at the tip and are captured and stabilized by MYO3A. The source of tip filaments is unknown, but the MYO15A-dependent elongation complex is a favored candidate. Short filaments along the shaft, nucleated by an unknown mechanism, are extended by new actin addition to produce stable core filaments. Additional short filaments form along the periphery as widening continues.

Reviewer #2 (Remarks to the Author):

Liao et al. describe a detailed procedure for peripheral widening of stereocilia. The authors used diverse imaging techniques, including SIM and expansion microscopy, to visualize nanoscale details of stereocilia and their changes. They identified a role for short actin filaments during the morphogenesis of stereocilia. Their work is very interesting and detailed and provides solid support for their finding. However, I have some questions about the imaging techniques used in this work, particularly expansion microscopy.

1. *The use of expansion microscopy in this work is limited: Figure 1c, Figure 7a, and Supplementary Figure 6b. The authors stated that they employed a protocol that is used for expanding parasites. In expansion microscopy, the expansion protocol has to be carefully optimized to ensure deformation-free expansion, especially tissue. In this work, the authors did not use expansion microscopy to measure the size of ultrasmall structures. Thus, expansion uniformity was not critical. However, providing an expansion factor would be a good indication of whether the expansion was truly uniform or not. If the expansion factor were smaller than the*

expansion factor of the gels without tissue, it would indicate that tissue expansion was not uniform. Therefore, can the authors please clarify the expansion factor in the expansion microscopy experiments and compare it with that of the black hydrogel, which does not contain any tissue?

We thank the reviewer for drawing our attention to this issue. Indeed, we found that the expansion factor differed (3.8x for the tissue and 3.9x for a gel without tissue), suggesting that expansion was very similar, but perhaps not perfectly uniform. We have added this information to the methods section to help readers better interpret our results. Specifically, we added the following sentences (Lines 672-674): “To estimate sample expansion parameters, tissue and gels were measured before and after expansion. Tissue expanded by a factor of 3.8x compared to 3.9x for gels without tissue.”

2. Figure 1b shows a top-down view of a clear ring structure, which was imaged using an Airyscan microscope. Such ring structures would be more clearly shown using expansion microscopy. However, such structures are not clearly visible in the 3D view of the expansion microscopy result shown in Figure 1c. Are such ring structures apparent in the expansion microscopy images?

The reviewer raises an excellent point. We revisited our images and replaced the expansion microscopy images in Figure 1c to show the ring structures. We appreciate the valuable feedback in refining our data presentation. The revised panels and legend are below.

Fig. 1: EGFP-actin incorporation pattern during stereocilia widening.

... c, P5 IHCs 18 hours post EGFP-actin transfection imaged by expansion microscopy, stained with antibodies against β -actin (ACTB) (magenta) and EGFP-actin (green). Panels show a 3D reconstruction (Scale bar represents 10 μ m), a x-z reslice through the center of stereocilia, and a top-down view of x-y slice (Scale bars represent 2 μ m). Yellow and blue arrows indicate row 1 and row 2, respectively...

3. Related to the above question, other experiments, such as the one shown in Figure 3f, could have benefited from the application of expansion microscopy. Why was expansion microscopy not used in these experiments requiring highly precise measurements?

We agree and we tried to visualize the localization of His-TMOD1 and His-CAPZ by expansion microscopy. However, the anti-His antibody failed to detect the probes after expansion. In Figure

7 we were able to detect His-TMOD1 with an anti-TMOD1 antibody, but the anti-CAPZ antibody did not detect His-CAPZ after expansion.

4. *In this study, the authors used Alexa 546 NHS ester to label all proteins in expansion microscopy. The use of NHS ester is widespread in expansion microscopy. However, none of the results included in this study show an Alexa 546 NHS ester channel. Is there any reason the Alexa 546 NHS ester channel is not shown?*

The NHS-ester data is presented in Figure 7 as an inverted greyscale image. In Figure 1, we did not stain with NHS ester and instead labeled with an anti-actin antibody.

5. *In the Methods section, the concentration of bisacrylamide is stated to be 2%. I believe this is incorrect, as the typical stock concentration of bisacrylamide is 2%. Please check the concentration and amend if not correct.*

The reviewer is correct. The final concentration of bisacrylamide was 0.1%, which we now indicate in the methods (Lines 651-652).

6. *I am unable to find a data availability statement and associated data in this manuscript. It would be useful to see the raw data. Please include a data availability statement.*

We agree that the raw data should be available for independent analysis. Raw image data will be deposited in Figshare as we resubmit the manuscript. Source data including measured values used will also be uploaded as spreadsheet that will be associated with this paper. We added a data availability statement to the manuscript (Lines 837-840).

Reviewer #3 (Remarks to the Author):

The mechanosensitive function of hair cells relies on stereocilia. However, the mechanisms underlying the lengthening and widening of stereocilia in inner ear hair cells remain an unresolved question in cell biology. In this paper, the authors focus on the widening of stereocilia and demonstrate that newly synthesized actin first incorporates at the stereocilia tip, followed by incorporation along the shaft, thereby promoting stereocilia widening. The authors also identified a previously undetected population of short actin filaments and proposed that these filaments act as intermediates in the assembly of new stereocilia core filaments. However, the reviewer has the following concerns regarding this work.

1. *Line 12: The research does not involve the vestibular system, and it is recommended that the term "vestibular system" be removed.*

This is a good point and we have removed the term "vestibular system".

2. *Line 80: In reference 5, the authors studied the incorporation of β -actin-GFP into stereocilia in organotypic cultures of hair cells from rat and mouse inner ear sensory epithelia, observing the incorporation and flux of actin, which results in the distribution of actin-GFP along the entire stereocilia. Given these findings, could the dynamic expression of EGFP-actin in the IHCs in the present study also be attributed to actin turnover?*

Our results largely replicate the findings in reference 5, but with the key difference that EGFP-actin surrounds the existing stereocilia core. We think this difference is due to improved resolution

and sensitivity of Airyscan, lattice SIM, and expansion microscopy. However, the Reviewer raises an excellent point that is relevant to the model we propose for stereocilia widening. Based on the data in this study, we suggest that short actin filaments are added to the sides of the existing core and then these filaments extend by monomer addition and/or by annealing with other growing filaments. If this model is correct, then an immature stereocilia core in the early days of widening could have relatively few long filaments that are surrounded by predominantly immature, short filaments. These short filaments may be assembling and disassembling according to actin concentration or the activity of binding proteins like Espin. In this case, EGFP-actin incorporation may be as much due to short filament turnover as filament addition or extension.

To address this point, and in response to question 4 below, we transfected P3 IHCs, as compared to P5-6 IHCs used in most of our experiments. Compared to later ages, the peripheral enrichment is less obvious in transfected P3 IHCs. Nevertheless, line scans across the width of transfected P3 stereocilia show a dip in EGFP-actin intensity in the middle, which is similar to what we see in P5 low expressors that remain relatively thin (compare Supplementary Figure 1d to 1b). The fact that P3 stereocilia are thinner than at P5 likely makes it harder to resolve peripheral incorporation. However, it could also be, as would be predicted by our model, that at younger ages fewer of the core filaments have fully matured and some are still capable of turnover, leaving only a narrow core of stable actin.

We added these data as a new Supplementary Figure 1c-d (shown below) and included the following summary in the results section (Lines 92-96): “We also transfected P3 IHCs, and in this case EGFP-actin incorporation was also peripheral, but less evident, suggesting the stable F-actin component of the core is thinner earlier in development (Supplementary Fig. 1c-d).”

Supplementary Fig. 1: EGFP-incorporation pattern in IHCs with prolonged transfection or at early postnatal age.

a, P5 IHCs 48 hours (2 DIV) after transfection with EGFP-actin (green and grey). F-actin is stained by phalloidin (magenta) to show stereocilia. **b**, Magnified insets from (**a**, yellow dashed boxes) showing stereocilia with higher or lower level of EGFP-actin (grey). Line scans drawn perpendicular to stereocilia shafts include the distribution of EGFP-actin (green) and F-actin (magenta) in transfected cells compared to F-actin (black dashed lines) in adjacent untransfected cells (untx). **c**, P3 IHCs 18 hours (1 DIV) after transfection with EGFP-actin (green). F-actin is stained by phalloidin (magenta). **d**, Magnified inset from (**c**, yellow dashed box) showing EGFP-actin (grey) and line scans drawn perpendicular to stereocilia shafts. Line graph includes the distribution of EGFP-actin (green) and F-actin (magenta) in transfected cells compared to F-actin (black dashed lines) in adjacent untransfected cells (untx). Scale bars represent 5 μ m. DIV: days in vitro.

3. How can the possibility be excluded that the observed actin incorporation results from anomalous actin overexpression?

Overexpression is an important caveat and a limitation of our study. In short, we were unable to devise an experiment to measure endogenous actin addition during widening. One challenge to doing this is the slow rate of widening, which is ~30 nm per day from P0-P8. Given that there are 9-13 nm between filaments (Song *et al. Journal of Structural Biology*. 2020. PMID: 31962158), we estimate that each day only 2-3 actin filaments are added to the periphery of a longitudinal section, which may be hard to detect even if we could distinguish newly synthesized endogenous actin from preexisting endogenous actin. To make this limitation clear, we added a “Limitations of study” paragraph at the end of the discussion as follows (Lines 339-344):

“Limitations of study

The primary limitation of this study is that the proposed widening mechanism partially relies on expression of exogenous actin and has not been validated through observation of endogenous actin incorporation. While our findings reveal a strong correlation of tip filaments to the occurrence of widening, the precise process by which actin adds to the periphery of core filaments and extends to the full length of stereocilia remains unclear.”

Nevertheless, there are reasons to favor the idea that EGFP-actin tracks and accentuates an endogenous mechanism. Importantly, EGFP-actin expression did not produce grossly distorted bundle morphologies. Instead, as is appropriate to the widening stage, the ectopic EGFP-actin increased stereocilia width without lengthening stereocilia, indicating the exogenous actin incorporated in much the same way as endogenous actin. In addition, we observed that EGFP-actin addition surrounded the otherwise stable stereocilia core, which is consistent with earlier work tracing endogenous proteins via multi-isotope imaging mass spectrometry (MIMS) (Zhang *et al. Nature*. 2012. PMID: 22246323). The MIMS study found that a large fraction of protein in developing stereocilia, which corresponded to the fraction of total protein that actin comprises, was stable and did not turnover. Although this study lacked the temporal or spatial resolution necessary to track actin addition at the stereocilia periphery, the overall stability of endogenous proteins is in line with our finding that EGFP-actin adds to the outside of an existing core.

4. Fig. 1b: Does the pattern of EGFP-actin incorporation into the stereociliary bundle depend on the age of the mice?

Yes, we expect that each stage of stereocilia development (for example, Stage III widening vs Stage IV elongation) requires a different pattern of actin incorporation to produce the particular

growth pattern. During the widening phase focused on in this study, there could be differences as stereocilia mature as detailed above in our response to question 2. However, it is also worth noting that the rate of widening is nearly linear between P0.5 and P8.5 (Krey *et al. PLoS Biology*. 2023. PMID: 37011103), suggesting a similar mechanism is likely used across this age range.

5.Line 83: What was the rationale for selecting P6 inner hair cells for transfection?

We selected P6 for several reasons. Dissection, culture, and injectoporation are optimal in our hands at P6. In addition, these stereocilia are wider and easier to image than at younger ages, yet stereocilia are widening at the same rate at P6 as at P3 when all of the experimental manipulations are far more difficult.

6.Lines 84-88: In conjunction with reference 5, hair cell cultures were transfected with β -actin-GFP. Forty-eight hours after transfection, the static cilia bundles were fully labeled. In contrast, in the present study, inner hair cells showed labeling at the tip and axis of the ciliary bundles only 18 hours post-transfection. Could this difference be due to the timing of transfection?

Thank you for raising this important point. It is indeed possible that a longer period of EGFP-actin expression would produce a different incorporation pattern. To address this point, we added new data from IHCs that were cultured for 48 hours after transfection. The stereocilia from these cells also have a peripheral pattern of EGFP-actin incorporation that is very similar to what we observed in cells analyzed 18 hours post transfection. These data are presented as in new supplementary Figure 1.

As an aside, we have sought to reconcile our observations with those published in reference 5. This has been challenging in the absence of source data owing to the vintage of the prior study. We speculate that the single example of an IHC 48 hours post transfection may have been particularly bright, making the peripheral shaft actin evident. Cells described in reference 5 imaged 12-24 hours after transfection seem to have a GFP-actin cap and no GFP-actin in the shaft. The shaft could have been too dim to image with the available technology, though we note that all the transfected cells are presented as composite images merged with phalloidin-stained F-actin. In our hands, merged images can obscure faint shaft labeling, for example in this expansion microscopy image from our original Figure 1:

7.Lines 106-107: According to what principle does the conclusion that "probe or G-actin is easily diffusible" arise?

We deleted this conclusion as it seemed to add little to our interpretation of the result.

8.Lines 106-110: When cells were extracted with saponin, the RPEL1-EGFP signal was drastically reduced to nearly undetectable levels. Under these conditions, JLA20 staining was enriched at the tips of row 2 stereocilia. What could explain this observation?

This is an interesting result suggesting that there are two populations of G-actin. One population is detected by the RPEL probe and is lost following saponin extraction and another that is anchored at row 2 stereocilia tips and detected after extraction by JLA20. The bound G-actin may not be able to bind the RPEL probe, as is the case with profilin-bound G-actin (Mouilleron *et al. EMBO J.* 2008. PMID: 19008859), thus explaining why it is not labeled by the RPEL probe.

9.Fig. 2a: How long after transfection of RPEL1-EGFP were the hair cells sampled for staining?

Thank you for pointing out this omission. Hair cells were samples 18 hours after transfection and we added this information to the figure legend.

10.Lines 118-119: Saponin can deplete the soluble G-actin pool in cells. Why, after saponin extraction, did the tip enrichment of TMOD1 signals become more pronounced, with a greater reduction in shaft signals? Additionally, how does this affect the signaling of F-actin?

In this experiment, EGFP-TMOD1 was overexpressed in the cell. As a result, we think there was a quantity of unbound EGFP-TMOD1 in stereocilia that was extracted. The remaining bound EGFP-TMOD1 was enriched at tips, which is consistent with the results of experiments where we probed permeabilized tissue with purified TMOD1 protein.

In the figure, fluorescent intensities were normalized to the maximal intensity of line scans drawn down the center of individual stereocilia to make it easier to compare tip-to-shaft ratios. However, the absolute intensity of TMOD1 was lower in extracted cells than unextracted cells, which is consistent with loss of unbound EGFP-TMOD1.

11.Lines 124-128: Are these tip filaments present exclusively at the tips of developing stereocilia? Additionally, are these filaments also found in the stereocilia of adult hair cells?

Tip filaments seem to be most enriched in developing stereocilia, but some fraction may be retained in mature stereocilia. Along these lines, we have preliminary data showing His-TMOD1 labeling of IHC stereocilia tips at P17 and at P23, although levels are low and nearing the limit of detectability (image below). Although these findings are potentially interesting, we have omitted them from the current manuscript because of the focus on the widening phase of growth.

12.Lines 147-148: How much time after the drug washout did the level of the pointed end recover significantly?

Pointed end recovery was assessed 4 hours after the drug was washed out. We added this information to the text (Line 151) and figure legend (Lines 405-406).

13.Fig. 4a, 4b: LatA, a drug that sequesters G-actin and depolymerizes dynamic F-actin structures, had no effect on the core actin filaments. What could explain this lack of effect?

This is an interesting question, and we think there are two elements at play. First, the duration of LatA treatment was relatively short, so stable F-actin structures may not disassemble enough to be readily detectable. Second, the stereocilia F-actin core is highly stable as revealed by multi-isotope imaging mass spectrometry of mouse pups fed N15 and by FRAP studies of stereocilia from transgenic mice expressing EGFP-actin from birth (Zhang *et al. Nature*. 2012. PMID: 22246323). In both cases, protein or EGFP-actin turnover is very low in the stereocilia shaft, likely because core filaments are highly cross-linked.

14.Fig. 5a: This panel illustrates that overexpression of actin leads to stereocilia widening and an increase in the number of tip filaments. However, there is no direct evidence establishing a link between the tip filaments and the observed stereocilia widening.

We agree that our data show a strong correlation of short actin filament levels and widening, but that this is not direct evidence of mechanism. To make this explicit to readers, we state this fact in the concluding "Limitations of study" paragraph (Line 340-344). Here, we also point out that tip filament levels also correlate well with developmental widening and that tip filaments rise and fall with overexpression and mutation of Myo3a, itself correlated with widening. Thus, the interpretation is based on multiple consistent correlations.

15.Fig. 6a: Compared with IHC transfected with EGFP-actin alone, IHC transfected with both EGFP-actin and RFP-AP-actin exhibit wider stereocilia bundles with greater incorporation of EGFP-actin. Could you provide an explanation for this observation?

In this case, we do not claim that EGFP-actin and RFP-AP-actin make stereocilia wider as we do not have evidence that co-transfected cells have significantly wider stereocilia than those expressing only EGFP-actin. In both conditions, there is marked variability in width. We do claim that RFP-AP-actin decreases EGFP-actin at tips but does not prevent it from incorporating along the shaft, where it may participate in widening.

To better demonstrate the effect of mutant RFP-actin and wild-type EGFP-actin expression, we included 6 hair cells in the figure instead of 4 and a magnified inset of the EGFP-actin channel to display the altered incorporation pattern of wild-type actin. We also edited the interpretation of the results as follows (Lines 180-184): "In contrast, EGFP-actin still incorporated similarly along the shaft whether or not it was co-expressed with mutant actin, which is evident in longitudinal line scans along stereocilia (Fig. 6c). These observations suggest that actin incorporation is regulated differently at the stereocilia tip than along the shaft."

16.Line 185: In addition to MYO3A/B, MYO15A, and many other proteins such as ESPIN, EPS8, and GPSM2 that regulate stereocilia elongation and widening, why were MYO3A/B and MYO15A specifically selected for regulatory analysis?

MYO3A/B attracted our attention because these tip-localized proteins have large effect on stereocilia width. In comparison, loss of the tip localized isoform of ESPIN (ESPN1) doesn't affect the width of auditory hair cell stereocilia (Ebrahim *et al. Nature Communications*. 2016. PMID: 26926603. Correspondingly, we did not see an increase in pointed end labeling when we transfected cells with ESPN1.

MYO15A was of particular interest because there is evidence that it can nucleate actin filaments, and it is required for normal stereocilia length. Along with MYO15A, EPS8, GPSM2, GNAI3, and Whirlin, are also required for normal length. However, ablating any member of that complex results in the same phenotype (short stereocilia), so we reasoned that it would be difficult to tease apart the contributions of particular members.

17.Lines 54, 55, 243-245: A prolonged transfection time may be necessary to observe the pattern of newly incorporated actin, which would help support the conclusion.

We appreciate this suggestion and have added data from inner hair cells 48 hours after transfection as detailed in our response to Question 2 above.

18. There is no direct evidence to suggest that the reduction of tip filaments in the stereociliary bundle influences its elongation and widening.

We consistently observe a correlation between tip filament levels and stereocilia shape in experiments where MYO3A/B or MYO15A are ablated. However, we have not devised a way to reduce tip filaments without simultaneously altering another factor that could influence the shape through a different mechanism. This is a limitation of our study.

19. As a general point, there are several datasets where the sample size (n) is not provided. Could the authors please ensure that the sample size (n) is included for all datasets, along with a clear explanation of what the number represents?

Thank you for making this important point. We have now added the sample size (n) used for statistical analysis in the figure legends, and whether it refers to stereocilia, cells, or cochleae is indicated.

20. Fig. 1b suggests that incorporation patterns vary at different time points following plasmid transfection. Please specify the time points at which samples were collected in all relevant results.

We collect all samples 18 hours post-transfection except the live imaging, which we now indicate in the figure legend.

Additional edits:

In the course of revising the manuscript, we elected to delete the following sentence related to Figure 7 panels g and i: "The mean fluorescence intensity at row 1 stereocilia tips, normalized to neighboring untransfected cells, was 2.0 ± 0.7 and 2.1 ± 0.7 (mean \pm SD) for EGFP-K50R-MYO3A and EGFP-MYO15A-2 expressing cells, respectively, a difference which was not statistically significant.". The reason for omitting these data is that we are unable to compare the relative expression of EGFP-MYO3A and EGFP-MYO15-2 as these experiments were conducted at different times with different imaging conditions. As a result, comparing His-TMOD1 levels may not reflect the relative ability of each myosin to increase tip filaments.